# Riemannian MeanFlow for One-Step Generation on Manifolds

**Zichen Zhong** [1]  **Haoliang Sun** [1]  **Yukun Zhao** [1]  **Yongshun Gong** [1]  **Yilong Yin** [1]

## Abstract

Flow Matching enables simulation-free training of generative models on Riemannian manifolds, yet sampling typically still relies on numerically integrating a probability-flow ODE. We propose Riemannian MeanFlow (RMF), extending Mean-Flow to manifold-valued generation where velocities lie in location-dependent tangent spaces. RMF defines an average-velocity field via parallel transport and derives a Riemannian MeanFlow identity that links average and instantaneous velocities for intrinsic supervision. We make this identity practical in a log-map tangent representation, avoiding trajectory simulation and heavy geometric computations. For stable optimization, we decompose the RMF objective into two terms and apply conflict-aware multi-task learning to mitigate gradient interference. RMF also supports conditional generation via classifier-free guidance. Experiments on spheres, tori, SO(3), and SE(3) demonstrate competitive one-step sampling with improved quality–efficiency trade-offs and substantially reduced sampling cost.

## 1. Introduction

Denoising generative models, including diffusion models (Ho et al., 2020; Song et al., 2021) and Flow Matching (FM) (Liu et al., 2023; Lipman et al., 2023), are increasingly extended beyond Euclidean spaces to address scientific and engineering tasks on structured, non-Euclidean domains (Uehara et al., 2025; Wen et al., 2025; Holderrieth et al., 2025). Flow Matching learns a time-dependent velocity field whose induced probability-flow ODE transports a base distribution to the data distribution. On Riemannian manifolds, Riemannian Flow Matching (RFM) retains key advantages such as simulation-free training and favorable scalability (Chen & Lipman, 2024). However, sampling typically still

---

[1]School of Software, Shandong University, Jinan, China. Correspondence to: Haoliang Sun <haolsun@sdu.edu.cn>.

*Proceedings of the 43rd International Conference on Machine Learning*, Seoul, South Korea. PMLR 306, 2026. Copyright 2026 by the author(s).

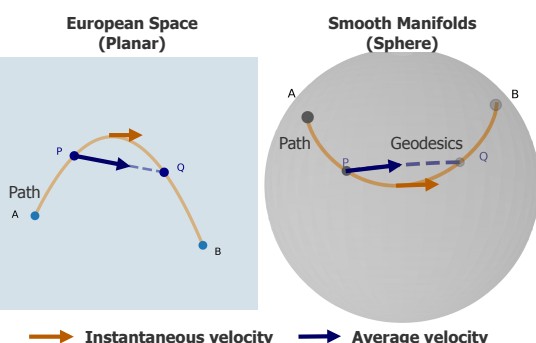

*Figure 1.* Instantaneous velocity (A→B) vs. average velocity (P→Q) on Euclidean straight lines and manifold geodesics.

requires numerically integrating the learned ODE on the manifold, which can be slow and may require many solver steps for high-quality samples; analogous iterative bottlenecks also arise in diffusion models.

A long line of work accelerates Euclidean sampling by learning *shortcuts* to multi-step samplers. Progressive distillation (Salimans & Ho, 2022) distills pretrained samplers into fewer steps. Consistency models (Song et al., 2023) enable one-stage training but often rely on carefully chosen schedules. Shortcut models (Frans et al., 2025), and IMM (Zhou et al., 2025) move toward end-to-end training and can achieve one-step denoising. MeanFlow (Geng et al., 2025) further parameterizes *long-range average velocity* through the MeanFlow identity, avoiding additional two-time self-consistency constraints and improving training stability. Recent unifications, such as Flow Map Matching (Boffi et al., 2025b) and $\alpha$-Flow (Zhang et al., 2026), connect these approaches under a common framework.

Despite this progress, extending MeanFlow-style generation to Riemannian manifolds is non-trivial. On a manifold, instantaneous velocities lie in point-dependent tangent spaces and must be compared under the Riemannian metric; as a result, even defining an "average velocity" requires mapping vectors to a common space, and naively reusing Euclidean identities breaks geometric consistency (Fig. 1).

To address these challenges, we propose **Riemannian MeanFlow (RMF)**, a fast generative modeling framework for Riemannian manifolds. RMF defines the average velocity over an interval by parallel-transporting instantaneous velocities along the trajectory to the tangent space at the

current state and averaging them there. Starting from this geometric definition, we derive a *Riemannian MeanFlow identity* that relates the average velocity to the instantaneous velocity via a covariant derivative along the path. To make this identity usable in practice, we map nearby manifold points to tangent displacements, so the required directional derivatives can be computed in a single Euclidean vector space without complex geometric computations or trajectory simulation.

RMF also introduces a stable optimization view of the resulting objective. We show that the RMF loss decomposes into two terms whose gradients can conflict in practice, and we cast the decomposition as a two-task learning problem with shared parameters (Caruana, 1997; Sener & Koltun, 2018). We adopt a lightweight conflict-aware update based on PCGrad (Yu et al., 2020) to mitigate gradient interference without manual schedule tuning. Finally, RMF supports conditional generation via classifier-free guidance, combining conditional and unconditional predictions in the common tangent space.

Contributions are summarized as

- We generalize MeanFlow to Riemannian manifolds by defining average velocity via parallel transport and deriving an intrinsic MeanFlow identity for supervision.

- We develop a practical, geometry-consistent training rule that operates in a common tangent space using logarithm maps, avoiding coordinate-based covariant computations and trajectory simulation.

- We improve optimization stability by casting the decomposed RMF objective as a two-task problem and applying PCGrad, and we further support conditional generation via classifier-free guidance.

- We evaluate RMF on spherical Earth/climate fields, protein structures on a flat torus, synthetic rotation on $SO(3)$, and robotic grasping dataset on $SE(3)$, achieving strong quality–efficiency trade-offs with substantially reduced sampling cost.

## 2. Preliminaries

**MeanFlow** (Geng et al., 2025) replaces the instantaneous velocity with an *average* velocity field, enabling one-step generation without iteratively solving an ODE at inference time. Formally, given target data $x_1 \sim p_1$ and a noise source $x_0 \sim p_0$, Flow Matching constructs a time-dependent interpolation path/flow $x_t = \sigma_t x_1 + \mu_t x_0$, $t \sim \mathcal{U}(0,1)$, where $\sigma_t$ and $\mu_t$ are predefined schedules (a common choice is $\sigma_t = 1 - t$ and $\mu_t = t$) (Lipman et al., 2024). Differentiating the path yields the (instantaneous) velocity along the trajectory, $v_t(x_t) = \frac{\mathrm{d}x_t}{\mathrm{d}t}$, which induces a probability path $\{p_t\}_{t \in [0,1]}$ to push the source distribution to the target distribution. Standard Flow Matching trains a neural network

to approximate the ground-truth velocity field and generates samples by numerically solving the corresponding ODE. To accelerate inference, MeanFlow defines the *average velocity* over an interval $[r, t]$ as

$$u(x_t, r, t) \;=\; \frac{1}{t-r} \int_r^t v_\tau(x_\tau)\, \mathrm{d}\tau, \qquad (1)$$

so that in principle a one-step generation is achieved when $r = 0$ and $t = 1$. Directly computing the ground-truth average velocity in Eq. (1) is intractable in general. MeanFlow therefore leverages an equivalent relation with the instantaneous velocity, obtained by differentiating the definition of $u$ with respect to $t$,

$$u(x_t, r, t) \;=\; v_t(x_t) - (t - r)\frac{\partial u(x_t, r, t)}{\partial t}, \qquad (2)$$

which provides a feasible training objective for learning the average-velocity field.

**Riemannian Flow Matching** (Chen & Lipman, 2024) extends Flow Matching to smooth manifolds. Let $(\mathcal{M}, g)$ be a smooth Riemannian manifold, where $g$ induces the geodesic distance, denote the tangent space at $x \in \mathcal{M}$ by $T_x \mathcal{M}$, and induces the Riemannian volume measure $d\mathrm{vol}_g$. We regard $p_0$ and $p_1$ as probability densities with respect to $d\mathrm{vol}$, where $p_1$ is the data distribution on $\mathcal{M}$ and $p_0$ is a simple base distribution on $\mathcal{M}$, commonly chosen as the uniform distribution on smooth manifolds. Analogous to the Euclidean case, consider a time-dependent flow $\psi_t : \mathcal{M} \to \mathcal{M}$ that induces a well defined probability path $\{p_t\}_{t \in [0,1]}$ on $\mathcal{M}$. Following RFM, probability paths are defined through conditional paths: $p_t(x) = \int p_t(x \mid x_1) q(x_1) d\mathrm{vol} x_1$, where $q(x_1) = p_1$, $p_t(x \mid x_1)$ are conditional paths. The path satisfies $p_0(\cdot \mid x_1) = p_0$, $p_1(\cdot \mid x_1) = p_1$. The flow is defined as the solution to the ODE $\frac{\mathrm{d}}{\mathrm{d}t}\psi_t(x) = v_t(\psi_t(x))$, where $v_t(\cdot) \in T_{\psi_t(\cdot)}\mathcal{M}$ is a time-dependent vector field. When geodesics admit closed-form expressions, a geodesic interpolation flow can be written via the exponential and logarithm maps as

$$x_t = \psi_t(x_0 | x_1) = \mathrm{Exp}_{x_1}\big(\kappa(t)\, \mathrm{Log}_{x_1}(x_0)\big), \qquad (3)$$

where $\psi_t(x_0 \mid x_1)$ denotes the point obtained by moving from $x_1$ toward $x_0$ along the geodesic according to $\kappa(t)$, and $\kappa : [0, 1] \to [0, 1]$ is monotonically decreasing with $\kappa(0) = 1$ and $\kappa(1) = 0$, and is typically set to $\kappa(t) = 1 - t$. A neural network $v_\theta$ is trained to approximate the ground-truth velocity field along the interpolation by minimizing

$$\mathcal{L}_{\mathrm{RFM}}(\theta) = \mathop{\mathbb{E}}_{t, x_0, x_1} \Big[ \big\| v_\theta(x_t) - v_t(\psi_t(x_0 \mid x_1)) \big\|_g^2 \Big], \quad (4)$$

where $t \sim \mathcal{U}(0, 1)$, $x_1 \sim p_1$, $x_0 \sim p_0$, $x_t = \psi_t(x_0 \mid x_1)$, and $\| \cdot \|_g$ denotes the norm induced by the Riemannian metric on the corresponding tangent space. At sampling

time, we generate samples by numerically integrating the learned probability-flow ODE $\frac{\mathrm{d}}{\mathrm{d}t} x_t = v_\theta(x_t)$ from an initial draw $x_0 \sim p_0$ to $t = 1$ using a standard ODE solver (e.g., Euler or Runge–Kutta (Hairer et al., 1993)).

## 3. Riemannian MeanFlow

### 3.1. Average Velocity on Riemannian Manifolds

On a Riemannian manifold $(\mathcal{M}, g)$, instantaneous velocities are tangent vectors that live in point-dependent spaces $T_{x_t}\mathcal{M}$. Consequently, a naive time average of velocities along a trajectory is ill-defined unless all vectors are first mapped to a common vector space. We therefore define the *average velocity* at time $t$ by parallel-transporting instantaneous velocities to $T_{x_t}\mathcal{M}$:

$$u(x_t, r, t) = \frac{1}{t - r} \int_r^t \mathcal{P}_{\tau \to t}^\gamma (v(x_\tau, \tau)) \, \mathrm{d}\tau, \quad (5)$$

where $r < t$ is a fixed reference time (independent of $t$), $\gamma : [r, t] \to \mathcal{M}$ is a smooth trajectory with $x_\tau = \gamma(\tau)$, and $v(x_\tau, \tau) \in T_{x_\tau}\mathcal{M}$ is the instantaneous velocity field along $\gamma$. The operator $\mathcal{P}_{\tau \to t}^\gamma : T_{x_\tau}\mathcal{M} \to T_{x_t}\mathcal{M}$ denotes parallel transport induced by the Levi–Civita connection along $\gamma|_{[\tau,t]}$, which makes the integral well-defined. When $\mathcal{M} = \mathbb{R}^d$ (Euclidean space), parallel transport reduces to the identity, and Eq. (5) recovers the standard Euclidean time average.

**From an intractable definition to a tractable identity.** Although Eq. (5) is geometrically natural, it is not directly usable as supervision: evaluating the integral requires explicit access to the entire trajectory segment $\{x_\tau\}_{\tau \in [r,t]}$ and parallel-transporting $v(x_\tau, \tau)$ for all $\tau$. To avoid trajectory simulation, we derive an identity that relates the average velocity to the instantaneous velocity through a covariant derivative at time $t$.

**Proposition 3.1** (Riemannian MeanFlow Identity). *Let $\gamma : [r, t] \to \mathcal{M}$ be a smooth trajectory with $x_\tau = \gamma(\tau)$ and $t > r$. Define $u(x_t, r, t) \in T_{x_t}\mathcal{M}$ by Eq. (5). If $v(\cdot, \tau)$ is sufficiently smooth along $\gamma$, then*

$$u(x_t, r, t) = v(x_t, t) - (t - r) \nabla_{\dot{\gamma}(t)} u(x_t, r, t). \quad (6)$$

*Proof sketch.* We first multiply both sides of Eq. (5) by $(t - r)$ then differentiate both sides with respect to $t$.

Here, $\dot{\gamma}(t) \in T_{x_t}\mathcal{M}$ is the trajectory velocity at time $t$. Since $u(\cdot)$ is a vector field along $\gamma$, the appropriate derivative is the covariant derivative $\nabla_{\dot{\gamma}(t)} u(x_t, r, t)$. Proposition 3.1 shows that it suffices to estimate the instantaneous velocity $v(x_t, t)$ and the local derivative term $\nabla_{\dot{\gamma}(t)} u(x_t, r, t)$; no time integration is required. For completeness, we provide the *standard local-coordinate expansion* of $\nabla_{\dot{\gamma}(t)} u(x_t, r, t)$

(i.e., the chain rule with Christoffel corrections) in Appendix A.

**Computing the derivative term without Christoffel symbols.** A direct coordinate implementation of $\nabla_{\dot{\gamma}(t)} u(x_t, r, t)$ requires manipulating a local basis and the associated Christoffel symbols, which is inconvenient in practice. Instead, we work in a common tangent space $T_{x_t}\mathcal{M}$ and represent nearby states via logarithmic maps $\mathrm{Log}_{x_t}(\cdot)$ (within a normal neighborhood). This avoids explicit parallel transport and enables consistent Euclidean computations in $T_{x_t}\mathcal{M}$.

We parameterize the interpolation path using Exp and Log (cf. Eq. (3)):

$$x_t = \mathrm{Exp}_{x_1}\big(\kappa(t) \, \mathrm{Log}_{x_1}(x_0)\big), \quad (7)$$

where $\kappa : [0, 1] \to [0, 1]$ is monotone decreasing with $\kappa(0) = 1$ and $\kappa(1) = 0$ (typically $\kappa(t) = 1 - t$). Closed-form expressions of Exp and Log for the manifolds considered in this work are summarized in Appendix C. Differentiating Eq. (7) gives the path velocity

$$\dot{x}_t := \frac{\mathrm{d}x_t}{\mathrm{d}t} = -\frac{\kappa'(t)}{\kappa(t)} \, \mathrm{Log}_{x_t}(x_1), \quad (8)$$

and in particular, when $\kappa(t) = 1 - t$,

$$\dot{x}_t = \frac{1}{1 - t} \, \mathrm{Log}_{x_t}(x_1). \quad (9)$$

For brevity, we denote the instantaneous velocity by $v(x_t, t) := \dot{x}_t$.

Next, we approximate the unknown average-velocity field by a neural network $u_\theta(x_t, r, t)$, i.e., $u(x_t, r, t) \approx u_\theta(x_t, r, t)$. Using Proposition 3.1, we form a computable target by replacing $\nabla_{\dot{\gamma}(t)} u(x_t, r, t)$ with the directional derivative of the network along the path:

$$u_{\mathrm{gt}}(x_t, r, t) := v(x_t, t) - (t - r) \left( \dot{x}_t \, \partial_{x_t} u_\theta + \partial_t u_\theta \right). \quad (10)$$

Here $\partial_{x_t} u_\theta$ and $\partial_t u_\theta$ denote Jacobian terms with respect to the input $x_t$ and time $t$, respectively, and the product $\dot{x}_t \, \partial_{x_t} u_\theta$ is interpreted as a Jacobian–vector product. In practice, we compute these quantities efficiently via Jacobian–vector products (JVPs), avoiding higher-order derivatives and coordinate-based covariant calculations.

### 3.2. A Multi-Task View of the Decomposed Objective

Given samples $(x_0, x_1, r, t)$ and the induced point $x_t$ on the interpolation path, RMF trains $u_\theta$ to regress the intrinsic average velocity $u(x_t, r, t)$ in Eq. (6) under the Riemannian metric $g$:

$$\mathcal{L}_{\mathrm{RMF}} = \mathbb{E}_{x_0, x_1, r, t} \left[ \| u_\theta(x_t, r, t) - u(x_t, r, t) \|_g^2 \right]. \quad (11)$$

Using the Riemannian MeanFlow identity in Eq. (6), $u(x_t, r, t) = v(x_t, t) - (t - r)\nabla_{\dot{\gamma}(t)} u(x_t, r, t)$, we can rewrite the squared error by expanding the cross term. This yields a decomposition analogous to the Euclidean case in (Zhang et al., 2026).

**Proposition 3.2** (Decomposed RMF Loss). *The RMF objective can be written as*

$$\mathcal{L}_{RMF} = \underbrace{\mathbb{E}_{x_0, x_1, r, t}\left[\|u_\theta(x_t, r, t) - v(x_t, t)\|_g^2\right]}_{\mathcal{L}_1(\theta)} + \quad (12)$$

$$\underbrace{2\,\mathbb{E}_{x_0, x_1, r, t}\left[\langle u_\theta(x_t, r, t), (t - r)\nabla_{\dot{\gamma}(t)} u(x_t, r, t)\rangle_g\right]}_{\mathcal{L}_2(\theta)} + C,$$

*where $C$ does not depend on $\theta$.*

**Practical form in a common tangent space.** As described in the previous subsection, we represent intermediate states in the common tangent space $T_{x_t}\mathcal{M}$ using $\mathrm{Log}_{x_t}(\cdot)$, which avoids coordinate-dependent Christoffel symbols and explicit parallel transport. Moreover, we approximate the instantaneous velocity by the network output at $r = t$, i.e., $v(x_t, t) \approx u_\theta(x_t, t, t)$. For the second term, we use a stop-gradient operator $\mathbf{sg}(\cdot)$ to prevent higher-order derivatives. Concretely, we use

$$\mathcal{L}_2(\theta) = 2\,\mathbb{E}_{x_0, x_1, r, t}\left[\langle u_\theta(x_t, r, t), (t - r)\,\mathbf{sg}(\xi_t)\rangle_g\right],$$

$$\xi_t := \dot{x}_t\,\partial_{x_t} u_\theta + \partial_t u_\theta, \quad (13)$$

where $\dot{x}_t = \frac{dx_t}{dt}$ is the path velocity and $\partial_{x_t} u_\theta, \partial_t u_\theta$ are computed efficiently via Jacobian–vector products (JVPs).

**Why a multi-task view?** Eq. (12) expresses RMF as the sum of two scalar objectives,

$$\mathcal{L}_{RMF}(\theta) = \mathcal{L}_1(\theta) + \mathcal{L}_2(\theta) + C, \quad (14)$$

which naturally defines a two-task learning problem with shared parameters $\theta$. In practice, the gradients $g_1 = \nabla_\theta \mathcal{L}_1(\theta)$ and $g_2 = \nabla_\theta \mathcal{L}_2(\theta)$ often exhibit negative cosine similarity (gradient conflict), causing oscillatory updates or one term dominating optimization, as also observed in Euclidean MeanFlow variants (Zhang et al., 2026). Rather than tuning manual weight strategies, we mitigate this conflict by operating directly on the task gradients in parameter space using PCGrad (Yu et al., 2020).

**PCGrad for two terms.** When $\langle g_1, g_2 \rangle < 0$, PCGrad removes the component of each gradient that conflicts with the other via orthogonal projection:

$$\tilde{g}_1 = g_1 - \mathbb{I}[\langle g_1, g_2 \rangle < 0]\frac{\langle g_1, g_2 \rangle}{\|g_2\|^2 + \varepsilon}\,g_2,$$

$$\tilde{g}_2 = g_2 - \mathbb{I}[\langle g_1, g_2 \rangle < 0]\frac{\langle g_2, g_1 \rangle}{\|g_1\|^2 + \varepsilon}\,g_1, \quad (15)$$

where $\varepsilon > 0$ is a small constant for numerical stability and $\mathbb{I}[\cdot]$ is the indicator function. The final update direction is

$$\tilde{g} = \tilde{g}_1 + \tilde{g}_2, \qquad \theta \leftarrow \theta - \eta\,\tilde{g}, \quad (16)$$

with learning rate $\eta$. Eq. (15) leaves gradients unchanged when they are aligned, and otherwise suppresses components that would increase the other loss to first order.

**Corollary 3.3** (PCGrad Guarantees for Intrinsic Manifold Losses). *Let $\theta \in \mathbb{R}^p$ and consider $\mathcal{L}(\theta) = \mathcal{L}_1(\theta) + \mathcal{L}_2(\theta)$, where each $\mathcal{L}_i : \mathbb{R}^p \to \mathbb{R}$ is a differentiable scalar function obtained from manifold-valued model outputs (e.g., via geodesic distances and smooth maps such as* Exp/Log *on a normal neighborhood). If the assumptions of Theorem 1 (resp. Theorem 2) in (Yu et al., 2020) hold for $\{\mathcal{L}_i\}$ as functions on $\mathbb{R}^p$ (e.g., smoothness/Lipschitz conditions on $\nabla_\theta \mathcal{L}$ and the stated convexity conditions), then the corresponding convergence and lower-bounded guarantees of PCGrad apply without modification.*

*Proof sketch.* PCGrad operates purely on Euclidean gradients $g_i = \nabla_\theta \mathcal{L}_i(\theta) \in \mathbb{R}^p$, and its analysis depends only on inner products and smoothness properties in parameter space. The manifold affects $g_i$ only through the chain rule in defining each scalar $\mathcal{L}_i(\theta)$; it does not alter the algebraic structure of the PCGrad updates or the inequalities used in (Yu et al., 2020). For the manifolds considered in this work, Exp, Log, and the metric-induced inner product are smooth within a normal neighborhood (away from standard singularities such as cut loci), ensuring differentiability of $\mathcal{L}_i(\theta)$ under the stated assumptions.

**Implementation.** We compute $g_1$ and $g_2$ via standard backpropagation on $\mathcal{L}_1$ and $\mathcal{L}_2$, apply Eq. (15) to obtain $\tilde{g}$, and pass $\tilde{g}$ to the optimizer. In our two-term setting, this adds only a few inner products per iteration and introduces no additional learnable parameters. The full procedure is summarized in Algorithm 1.

### 3.3. Classifier-Free Guidance (CFG)

We extend RMF to conditional generation via classifier-free guidance. Let $c$ denote a class label (or, more generally, a conditioning signal). We train a single network $u_\theta$ that supports both conditional and unconditional predictions by randomly dropping the condition during training: with probability $p_{\mathrm{drop}}$ we replace $c$ by a null token $\varnothing$. We write the resulting model as $u_\theta(x_t, r, t|c)$, where $c \in \mathcal{C} \cup \{\varnothing\}$. We apply the same guidance rule to the instantaneous velocity target $v(x_t, t|c)$.

With the decomposed RMF objective in Eq. (12), the corre-

**Algorithm 1** Training Riemannian MeanFlow

1: **Require:** $p_1$, $p_0$, schedule $\kappa(t)$, network $u_\theta$, optimizer Opt, $\varepsilon > 0$
2: **repeat**
3:     Sample minibatch $x_1 \sim p_1$, $x_0 \sim p_0$, and $(r, t)$ with $0 \leq r < t \leq 1$
4:     $x_t \leftarrow \mathrm{Exp}_{x_1}\big(\kappa(t) \mathrm{Log}_{x_1}(x_0)\big)$
5:     $\dot{x}_t \leftarrow -\frac{\kappa'(t)}{\kappa(t)} \mathrm{Log}_{x_t}(x_1)$
6:     $(u, \xi_t) \leftarrow \mathrm{JVPs}(u_\theta, (x_t, r, t), (\dot{x}_t, 0, 1))$
7:     $\xi_t \leftarrow \mathbf{sg}(\xi_t)$
8:     $\mathcal{L}_1 \leftarrow \|u - \dot{x}_t\|_g^2$
9:     $\mathcal{L}_2 \leftarrow 2 \langle u, (t - r)\xi_t \rangle_g$
10:    $\mathrm{g}_1 \leftarrow \nabla_\theta \mathcal{L}_1$, $\mathrm{g}_2 \leftarrow \nabla_\theta \mathcal{L}_2$
11:    $\tilde{\mathrm{g}} \leftarrow \mathrm{PCGrad}(\mathrm{g}_1, \mathrm{g}_2; \varepsilon)$
12:    $\theta \leftarrow \mathrm{Opt}\big(\theta, \tilde{\mathrm{g}}\big)$
13: **until** end of training

sponding CFG training loss is

$$\mathcal{L}_{\mathrm{CFG}}(\theta) = \mathbb{E}_{x_0, x_1, r, t}\Big[\|u_\theta(x_t, r, t|c) - v(x_t, t|c)\|_g^2\Big] +$$

$$2\,\mathbb{E}_{x_0, x_1, r, t}\Big[\big\langle u_\theta(x_t, r, t|c), (t - r)\,\mathbf{sg}\big(\xi_t(c)\big)\big\rangle_g\Big], \tag{17}$$

where $\xi_t(c) := \dot{x}_t \partial_{x_t} u_\theta(\cdot|c) + \partial_t u_\theta(\cdot|c)$. In practice, the instantaneous velocity target is independent of the conditioning signal. We can also compute $\xi_t(c)$ efficiently via a JVP with direction $(\dot{x}_t, 0, 1)$, and apply $\mathbf{sg}$ to avoid higher-order differentiation, exactly as in Eq. (13).

## 4. Related Work

**Riemannian Denoising Generative Models**. Motivated by the success of diffusion and score-based models in Euclidean spaces (Ho et al., 2020; Song et al., 2021), several works (Bortoli et al., 2022; Huang et al., 2022; Lou et al., 2023; JO & Hwang, 2024) extend diffusion modeling to data supported on Riemannian manifolds. Compared to the Euclidean setting, manifold diffusions are typically formulated via SDEs driven by Brownian motion on the manifold, whose transition density (i.e., the heat kernel) is generally intractable; consequently, training often relies on discretized forward simulations and/or approximations to the conditional score (Chen & Lipman, 2024). Some methods approximate the conditional score explicitly, while others adopt implicit score matching objectives (Hyvärinen & Dayan, 2005) that avoid explicit score targets but require estimating divergence/trace terms, which can increase computation and introduce high-variance gradients, hindering scalability with large neural networks (Chen & Lipman, 2024). In contrast, motivated by simulation-free objectives for continuous normalizing flows on manifolds (Rozen et al., 2021; Ben-Hamu et al., 2022), Riemannian Flow Matching (Chen & Lipman,

2024) learns conditional velocity fields using closed-form geodesic constructions, enabling simulation-free training on common manifolds (e.g., spheres, tori, and SO(3)).

**One-Step Diffusion/Flow Models**. While diffusion models and flow matching have achieved remarkable success in image generation, a common criticism is that producing high-quality samples often requires many iterative steps. In Euclidean spaces, early work accelerated sampling via a two-stage pipeline that distills a pretrained multi-step sampler into a few-step generator (Salimans & Ho, 2022). Building on this line, consistency models (Song et al., 2023) made it possible to train few-step generative models from scratch in a single stage, without relying on a teacher. *Shortcut models* (Frans et al., 2025) and Inductive Moment Matching (IMM) (Zhou et al., 2025) further improve upon consistency models through *end-to-end* training, and can achieve one-step denoising with a single network. MeanFlow (Geng et al., 2025) introduces the MeanFlow identity and directly parameterizes the long-range average velocity, eliminating the additional two-time self-consistency constraints used in *Shortcut* and IMM. This yields more stable optimization and substantially narrows the gap between few-step and multi-step models trained from scratch. More recently, Flow Map Matching (FMM) (Boffi et al., 2025a;b) and $\alpha$-Flow (Zhang et al., 2026) propose unifying formulations that subsume consistency trajectory models (CTM) (Kim et al., 2024), *Shortcut* models, and MeanFlow as special cases. Recently, Riemannian Consistency Models (Cheng et al., 2025) extend few-step consistency modeling to non-Euclidean settings while respecting the intrinsic constraints imposed by Riemannian geometry. Alternatively, Generalised Flow Maps (GFM) (Davis et al., 2026) generalize Flow Map Matching to Riemannian manifolds, yielding a new class of few-step geometric generative models.

Among recent unifying one-/few-step formulations, GFM and $\alpha$-Flow are closest to our setting; we therefore clarify how RMF differs. In contrast to GFM, which extends Flow Map Matching by learning flow maps between arbitrary time pairs on manifolds, *RMF is a direct Riemannian generalization of MeanFlow that parameterizes long-range dynamics through an intrinsic average-velocity field defined with geometry-aware operators (e.g., parallel transport).* Moreover, unlike Euclidean $\alpha$-Flow, which stabilizes the decomposed objective via a manually specified, iteration-dependent weighting schedule, *RMF treats the decomposed terms as a two-task objective and mitigates their gradient conflicts through conflict-aware multi-task optimization, avoiding additional schedule tuning.*

A concurrent independent work (Woo et al., 2026) shares our broad objective of extending average-velocity-based MeanFlow generation to Riemannian manifolds. However, the two studies *differ substantially in their objective con-*

*struction and optimization strategies.* Specifically, Woo et al. adopt a flow map perspective to derive three equivalent representations of the average velocity (Eulerian, Lagrangian, and semigroup identities), each yielding a distinct training objective, whereas our work defines the average velocity through parallel transport of instantaneous velocities along the trajectory to a common tangent space, deriving an intrinsic Riemannian MeanFlow identity that relates average and instantaneous velocities via covariant derivatives. For optimization, Woo et al. employ $x_1$-prediction parameterization, refined time sampling distributions, and adaptive loss weighting to enhance model stability, whereas we decompose the objective function into two terms and formulate it as a multi-task learning problem, leveraging PCGrad to mitigate gradient conflicts between the decomposed losses.

# 5. Experiments

We evaluate RMF on multiple non-Euclidean datasets spanning diverse manifold domains, including a flat torus and synthetic rotations on SO(3). Concretely, we consider (i) the spherical Earth dataset, (ii) torus protein and RNA torsion-angle datasets, (iii) a synthetic SO(3) dataset, and (iv) robotic grasping dataset on SE(3). In experiments, we seek to answer the following questions:

1. Do the decomposed loss terms exhibit gradient conflicts in practice, and does conflict-aware optimization improve performance?

2. Can RMF achieve competitive one-step generation quality across diverse manifolds?

3. How does classifier-free guidance behave on manifolds, and does RMF support effective conditional sampling?

4. Does RMF scale to high-dimensional manifolds, and how does its performance vary with intrinsic dimension?

**Experimental setup.** All datasets are split into train/val/test = 8/1/1. We follow a strict evaluation protocol: models are trained on the training set, hyperparameters are selected on the validation set, and the final results are reported on the test set using the selected model. We report results for two variants: RMF and RMF-MT, where RMF-MT applies conflict-aware multi-task optimization to the decomposed objective in Section 3.2. $\kappa(t) = 1 - t$. Detailed hyperparameters for all settings are provided in Appendix I.

**Evaluation metric.** To quantify the discrepancy between one-step samples (1 NFE) and the test distribution, we report the Maximum Mean Discrepancy (MMD) between generated samples and held-out test data. Following (Gretton et al., 2012), MMD admits unbiased (U-statistic) and biased

(V-statistic) estimators; we use the V-statistic form:

$$\text{MMD}^2(X, Y) = \frac{1}{n^2} \sum_{i,j=1}^{n} k(x_i, x_j) + k(y_i, y_j) - 2k(x_i, y_j),$$

where $k(x, y) := \exp\left(-\lambda \, d_g^2(x, y)\right)$ is an RBF kernel defined using the squared geodesic distance $d_g^2(\cdot, \cdot)$ on $\mathcal{M}$, and $\lambda = 1$ is the bandwidth parameter. Additional implementation details are provided in Appendix I.

**Baselines.** We compare RMF against state-of-the-art geometric generative models, including Riemannian Flow Matching (RFM) (Chen & Lipman, 2024), Riemannian Consistency Models (RCM) (Cheng et al., 2025) using its directly trained variant Riemannian Consistency Training (RCT; no distillation), and Generalized Flow Maps (GFM) with its three reported variants (G-PSD/G-ESD/G-LSD) (Davis et al., 2026). *Among the GFM variants, G-LSD consistently performs best, substantially outperforming G-PSD and G-ESD. Our RMF achieves performance comparable to G-LSD, the strongest baseline.*

## 5.1. Gradient Conflict Analysis of Loss

We empirically verify the gradient conflicts on geometric data between the two loss terms introduced in Section 3.2. We quantify gradient interference using the cosine similarity $\cos(\nabla \mathcal{L}_1(\theta), \nabla \mathcal{L}_2(\theta))$, where negative values indicate conflicting update directions. Figure 2 plots the per-iteration cosine similarity and its running average on the Earth dataset (full results for all datasets are provided in Appendix H). Across categories, we observe frequent negative cosine similarities, indicating that the two objectives often produce incompatible gradients during training.

This observation is consistent with the gains of conflict-aware optimization in Table 1: RMF-MT yields larger improvements over RMF on categories where the average cosine similarity is more negative (e.g., *Flood*), while the improvement is smaller when the gradients are relatively less conflicting (e.g., *Volcano*). Overall, the results suggest that explicitly mitigating gradient conflicts is most beneficial in settings where the decomposed terms exhibit stronger and more persistent disagreement.

## 5.2. Comparisons to Baselines

**Sphere.** We evaluate RMF on the Earth disaster dataset defined on the 2D sphere $\mathbb{S}^2$. The dataset is introduced in (Mathieu & Nickel, 2020) and curated from multiple public sources (NOAA, 2020a;b; Brakenridge, 2017; EOSDIS, 2020). Results are summarized in Table 1 using MMD at 1 NFE. RMF attains the best performance on *Volcano* (0.092), outperforming all baselines by a clear margin, while RMF-MT achieves the best result on *Flood* (0.048). On *Earthquake* and *Fire*, the strongest baseline G-LSD remains the

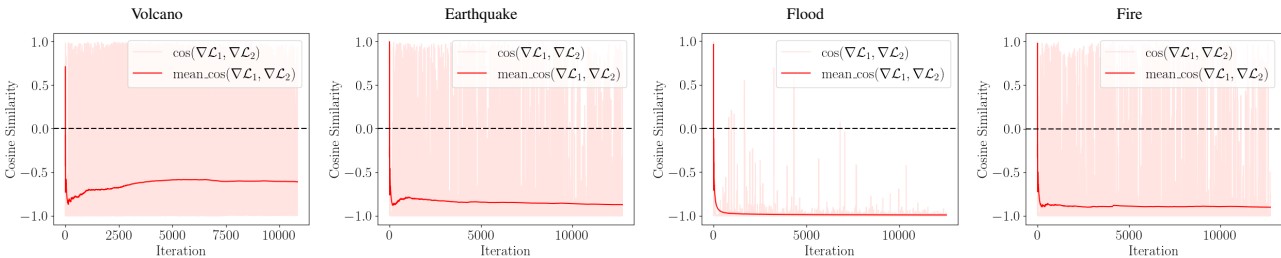

*Figure 2.* Cosine similarity between full-parameter gradients $\nabla_\theta \mathcal{L}_1$ and $\nabla \theta \mathcal{L}_2$ across Earth categories during training (per-iteration values and running average); negative values indicate gradient conflicts.

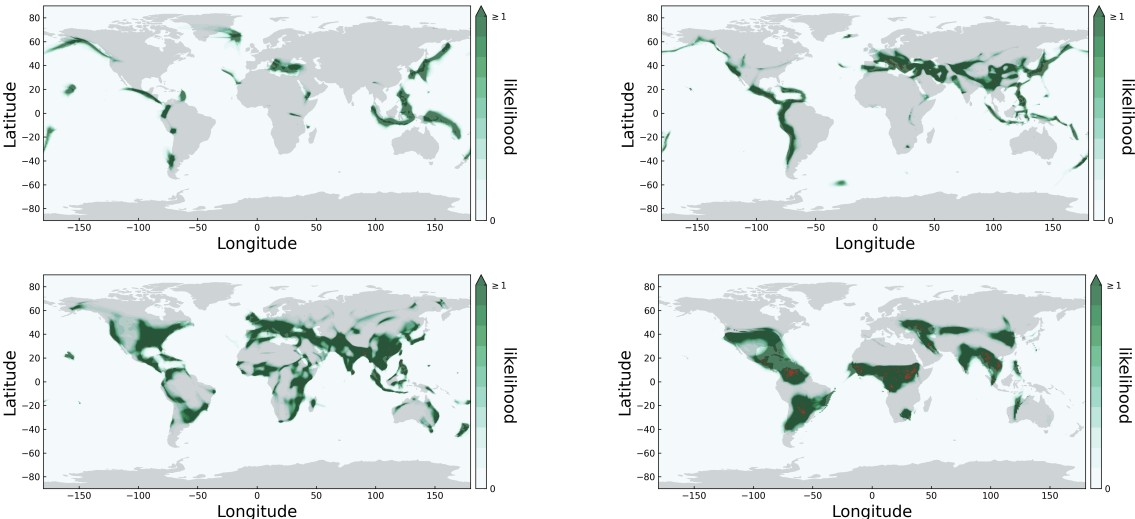

*Figure 3.* Earth dataset: generated sample distributions overlaid with ground-truth test data (red). Top-left: Volcano; top-right: Earthquake; bottom-left: Flood; bottom-right: Fire.

*Table 1.* MMD ($\downarrow$) at 1 NFE on the Spherical test dataset. Values are averaged over 5 runs with different random seeds. Best is **bold**; second best is underlined.

|  | Volcano | Earthquake | Flood | Fire |
|---|---|---|---|---|
| Dataset size | 827 | 6,120 | 4,875 | 12,809 |
| RFM | 0.351 | 0.309 | 0.272 | 0.377 |
| RCT | 0.155 | 0.053 | 0.086 | 0.080 |
| G-PSD | 0.139 | 0.045 | 0.067 | 0.046 |
| G-ESD | 0.184 | 0.178 | 0.178 | 0.196 |
| G-LSD | 0.115 | **0.032** | 0.065 | **0.027** |
| **RMF** | **0.092** | 0.042 | 0.068 | 0.042 |
| **RMF-MT** | 0.102 | 0.035 | **0.048** | 0.032 |

top performer (0.032 and 0.027), and RMF-MT achieves the second-best results (0.035 and 0.032), improving over RMF (0.042 on both). Overall, conflict-aware multi-task optimization is consistently beneficial on Earthquake/Flood/Fire, while it slightly degrades on Volcano, likely due to the much smaller dataset size (827 samples). Qualitative visu-

alizations are provided in Figure 3. Low-data regime with statistical tests are reported in Table 16 and Table 17 in Appendix G. We report KLD and NLL score in Table 10 and Table 12.

**Torus.** We evaluate RMF on torsion-angle data, where periodic variables induce a torus geometry. Specifically, we use four 2D protein subsets from (Lovell et al., 2003) (General, Glycine, Proline, and PrePro) and a 7D RNA backbone torsion dataset from (Murray et al., 2003). Table 2 reports MMD at 1 NFE. On protein subsets, RCT only performs best on the *General* set, while both RMF-MT and G-LSD are best on the remaining subsets. RMF-MT consistently improves over RMF and matches the best baseline on *Glycine/Proline/PrePro*. On the higher-dimensional *RNA (7D)* dataset, RMF-MT achieves the best overall performance, outperforming all baselines. Qualitative visualizations are provided in Appendix H. Low-data regime with statistical tests are reported in Table 16 in Appendix G. We report NLL score in Table 11.

**SO(3).** We evaluate RMF on four synthetic SO(3) datasets from (Cheng et al., 2025). Quantitative results are reported

*Table 2.* MMD ($\downarrow$) at 1 NFE on Torus test datasets (mean over 5 runs with different random seeds). Best is **bold**; second best is underlined.

|          | General | Glycine | Proline | PrePro | RNA   |
| -------- | ------- | ------- | ------- | ------ | ----- |
| Dataset size | 138,208 | 13,283 | 7,634 | 6,910 | 9,478 |
| RFM      | 0.45    | 0.27    | 0.52    | 0.47   | 0.68  |
| RCT      | **0.01** | 0.04 | 0.05 | 0.06 | 0.11  |
| G-PSD    | 0.11    | 0.05    | 0.07    | 0.08   | 0.14  |
| G-ESD    | 0.29    | 0.13    | 0.44    | 0.26   | 0.45  |
| G-LSD    | 0.02 | **0.03** | **0.04** | **0.05** | 0.08 |
| **RMF**  | 0.11    | 0.04 | 0.09    | 0.09   | 0.20  |
| **RMF-MT** | 0.04  | **0.03** | **0.04** | **0.05** | **0.07** |

*Table 3.* MMD ($\downarrow$) at 1 NFE on the $SO(3)$ test set (mean over 5 runs with different random seeds). Best is **bold**; second best is underlined.

|          | Cone  | Fisher | Line  | Swiss Roll |
| -------- | ----- | ------ | ----- | ---------- |
| Dateset size | 20k | 40k  | 40k   | 40k        |
| RFM      | 0.311 | 0.154  | 0.095 | 0.346      |
| RCT      | 0.096 | 0.071 | 0.051 | 0.082 |
| G-LSD    | **0.044** | 0.073 | 0.037 | **0.032** |
| **RMF**  | 0.057 | 0.076  | 0.038 | 0.080      |
| **RMF-MT** | 0.049 | **0.039** | **0.035** | 0.061 |

in Table 3 and qualitative samples are provided in Appendix H. RMF-MT achieves the best MMD on *Fisher* and *Line*, and attains the second-best performance on *Cone* and *Swiss Roll*, where G-LSD is the strongest baseline. Across all four datasets, RMF-MT consistently improves over RMF, indicating that conflict-aware optimization is beneficial for learning accurate distributions on $SO(3)$. Overall, these results, together with the sphere and torus experiments, demonstrate that RMF provides strong one-step generation performance across diverse manifold domains. We conduct multiple sampling on the sphere, torus and $SO(3)$ dataset and report MMD ($\downarrow$) in Figure 7 in Appendix G.

**SE(3).** We further evaluate RMF on an $SE(3)$ dataset derived from parallel-jaw gripper grasping experiments (Eppner et al., 2020), where each sample represents a rigid-body grasp pose. The results are reported in Table 4. RMF achieves the highest grasping success rate in the low-step regime (1-step and 2-step) and remains competitive at larger sampling steps, matching the best result at 7-step, demonstrating its effectiveness for robotic grasp generation on $SE(3)$. Interestingly, RMF-MT achieves a lower success rate than RMF. One possible explanation is that RMF-MT may better fit the overall $SE(3)$ pose distribution, while grasp success additionally depends on physical feasibility and collision avoidance. Thus, better distribution fitting does not necessarily translate into higher grasping success.

*Table 4.* Grasping success rate% ($\uparrow$) on the $SE(3)$ dataset across different sampling steps. Best is **bold**; second best is underlined. *Notice: The metric is the grasping success rate, not MMD.*

| Step     | 1   | 2   | 3   | 4   | 5   | 6   | 7   |
| -------- | --- | --- | --- | --- | --- | --- | --- |
| Dataset size |     |     |  15.6M |  |     |     |     |
| RFM      | 3.2 | 23  | 38  | 59  | 80 | 82 | 88 |
| G-LSD    | 60 | 75 | 81 | **88** | 88 | 87 | **90** |
| RMF      | **65** | **80** | **82** | 86 | 88 | 88 | **90** |
| RMF-MT   | 60 | 67 | 70 | 72 | 75 | 78 | 81 |

*Table 5.* MMD ($\downarrow$) at 1 NFE on the $SO(3)$ mixture test set.

|        | Cone  | Fisher | Swiss | Mixture |
| ------ | ----- | ------ | ----- | ------- |
| uncond | -     | -      | -     | 0.439   |
| CFG    | 0.115 | 0.159  | 0.546 | **0.234** |

### 5.3. Ablation Study

**Classifier-Free Guidance.** We conduct a preliminary study of classifier-free guidance (CFG) for manifold-valued generation. We build a labeled $SO(3)$ mixture dataset by uniformly mixing *Cone*, *Fisher*, and *Swiss Roll*, each with 20k samples (60k total). We train an unconditional model (*uncond*) and a CFG-capable conditional model on the same data. We then generate samples conditioned on each label and evaluate MMD to the corresponding component, and also to the overall mixture. Results are reported in Table 5 with visualizations in Figure 4.

Although per-class MMD remains relatively high, CFG demonstrates clear label-dependent control that the unconditional model lacks. In Figure 4, the unconditional model produces largely label-insensitive samples, whereas CFG recovers distinct geometric structures: the *Cone* condition yields a clear loop, and the *Swiss Roll* condition captures the component's characteristic spatial organization. The generated samples under different labels are also visually well-separated, indicating that guidance effectively selects different modes rather than collapsing to an averaged mixture. Consistently, CFG also improves mixture-level fidelity, reducing MMD to the mixture from 0.439 (*uncond*) to 0.234. Overall, these results support that CFG can effectively steer generation on $SO(3)$, even in the challenging 1-step setting.

**High-Dimensional Manifold Scalability.** To assess whether RMF remains effective on high-dimensional Riemannian manifolds, we conduct an ablation study on a synthetic $\mathbb{S}^{d-1}$ hypersphere dataset. When $d = 3$, the data reduces to a circular ring, while larger $d$ yields progressively higher-dimensional spherical geometry. We evaluate generation quality using MMD between 1 NFE and the test set. Table 6 reports "MMD / epoch", where the first number is the achieved MMD ($\downarrow$) and the second is the epoch at

*Table 6.* MMD ($\downarrow$) for 1 NFE on hyperspheres $\mathbb{S}^{d-1}$, with epochs to reach each score.

| MMD/epoch | 3 | 4 | 8 | 16 | 32 | 64 | 128 |
|---|---|---|---|---|---|---|---|
| EMF | 0.193 / 50 | 0.154 / 32 | 0.108 / 60 | 0.115 / 51 | 0.150 / 50 | 0.159 / 132 | 0.357 / > 400 |
| RFM | 0.088 / 18 | 0.080 / 16 | 0.073 / 2 | 0.054 / 2 | 0.040 / 2 | 0.037 / 2 | 0.031 / 2 |
| **RMF** | **0.066** / 18 | **0.056** / 10 | **0.055** / 6 | **0.038** / 6 | **0.035** / 6 | **0.028** / 4 | **0.026** / 6 |

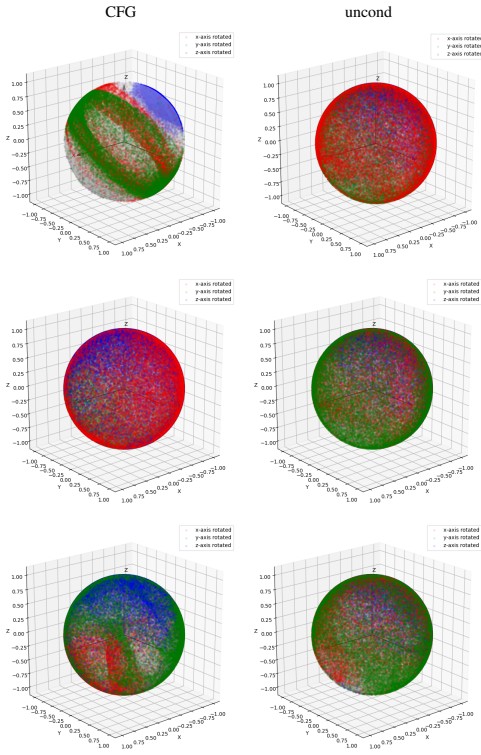

*Figure 4.* Visualization of samples generated by CFG and unconditional models.

which the method reaches that MMD. Across all dimensions, RMF consistently attains the lowest MMD among the compared methods and does so with a small number of epochs. The advantage becomes more pronounced as dimensionality increases: for $d = 64, 128$, Euclidean Mean-Flow (EMF) degrades substantially, whereas RMF remains stable and achieves strong results. RFM also performs well on this simple benchmark, but RMF is consistently better. Overall, these results indicate that RMF scales favorably with manifold dimension.

We evaluate RMF on a real-world high-dimensional promoter DNA sequence dataset from FANTOM5 (Hon et al., 2017), where each sample has dimension $1024 \times 4$. As shown in Table 7, RMF achieves an MSE of 0.030 and a k-mer correlation of 0.94 using only a single NFE. Its MSE matches Fisher FM (Davis et al., 2024) and Eulerian RMF (Woo et al., 2026), while outperforming Dirichlet FM (Stark et al., 2024); importantly, RMF requires far fewer

*Table 7.* Promoter DNA sequence generation results. Best is **bold**; second best is underlined.

| Method | NFE | MSE($\downarrow$) | k-mer corr. ($\uparrow$) |
|---|---|---|---|
| Dirichlet FM | 100 | 0.034 | N/A |
| Fisher FM | 100 | 0.030 | **0.96** |
| Eulerian RMF | 1 | 0.030 | **0.96** |
| Lagrangian RMF | 1 | **0.027** | 0.88 |
| Semigroup RMF | 1 | 0.030 | 0.84 |
| **RMF(ours)** | 1 | 0.030 | 0.94 |

sampling steps than the FM baselines. Compared with the RMF variants of Woo et al. (Woo et al., 2026), our method achieves competitive overall performance and preserves strong sequence-level statistical fidelity, as reflected by its high k-mer correlation.

## 6. Conclusion

We proposed a one-step generative framework for data supported on Riemannian manifolds. RMF defines an intrinsic average-velocity field in point-dependent tangent spaces and derives a Riemannian MeanFlow identity that provides tractable, manifold-consistent supervision in a log-map tangent-space representation, avoiding trajectory simulation and geometric computations. RMF further supports classifier-free guidance for conditional generation, and we introduce RMF-MT to mitigate gradient conflicts in the decomposed objective via conflict-aware multi-task optimization. Empirically, across diverse manifold domains (sphere, flat torus, SO(3) rotations, and SE(3)), RMF achieves favorable quality–efficiency trade-offs, matching the test distribution with low MMD while substantially reducing sampling cost compared to multi-step geometric baselines. Future work includes extending RMF to broader geometric settings where closed-form geodesics are unavailable.

## Acknowledgment

The work is supported in part by Natural Science Foundation of China (No. U23A20389, 62476154), Natural Science Foundation of Shandong Province (No. ZR2024MF101, ZR2024ZD03), and Young Expert of Taishan Scholars (No. tsqn202312026).

## Impact Statement

RMF improves sampling efficiency for manifold-valued generative modeling, which can accelerate scientific workflows involving spherical, toroidal, SO(3), and SE(3) data. Risks include misuse of synthetic samples and over-reliance on generated outputs, especially in high-stakes domains (e.g., disaster-related data). We recommend treating RMF as a research tool, clearly labeling synthetic content, and applying domain constraints and expert validation before any real-world use.

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

## A. Riemannian MeanFlow in Local Coordinates

This section provides a coordinate-level definition of the covariant derivative $\nabla_{\dot{\gamma}(t)} u(x_t, r, t)$ used in Proposition 3.1. Throughout, let $\dim(\mathcal{M}) = d$ and let $\gamma$ be a smooth curve on $\mathcal{M}$ with $x_t = \gamma(t)$.

**Local coordinates and notation.** Fix a chart $(x^1, \ldots, x^d)$ around $x_t$. The coordinate basis of the tangent space is $\{\partial_1, \ldots, \partial_d\}$, where $\partial_k := \frac{\partial}{\partial x^k}$. A tangent vector field $u(\cdot, r, t)$ along $\gamma$ can be written as

$$u(x_t, r, t) = u^k(x_t, r, t)\, \partial_k\big|_{x_t}, \qquad \dot{x}^i(t) = \frac{dx^i(t)}{dt}.$$

Here, $k \in \{1, \ldots, d\}$ indexes the coordinate components of $u$ in the chosen basis, and repeated indices are summed over $1, \ldots, d$ (Einstein summation convention). The notation $\left(\frac{D}{dt} u\right)^k$ denotes the $k$-th coordinate component of $\frac{D}{dt} u$ in the basis $\{\partial_k\}$.

**Covariant time derivative along $\gamma$.** By definition, the covariant derivative of $u$ along $\gamma$ is the covariant time derivative

$$\frac{D}{dt} u(x_t, r, t) = \nabla_{\dot{\gamma}(t)} u(\cdot, r, t)\Big|_{x_t}.$$

In local coordinates, it expands as

$$\left(\frac{D}{dt} u(x_t, r, t)\right)^k = \frac{d}{dt} u^k(x_t, r, t) + \Gamma^k_{ij}(x_t)\, \dot{x}^i(t)\, u^j(x_t, r, t), \tag{18}$$

or equivalently, in vector form,

$$\frac{D}{dt} u(x_t, r, t) = \left(\frac{d}{dt} u^k(x_t, r, t) + \Gamma^k_{ij}(x_t)\, \dot{x}^i(t)\, u^j(x_t, r, t)\right) \partial_k\Big|_{x_t}, \tag{19}$$

where $\Gamma^k_{ij}$ are the Christoffel symbols of the Levi–Civita connection in this chart.

**Chain rule for $\frac{d}{dt} u^k(x_t, r, t)$.** Treat each component $u^k$ as a scalar function of $(x_t, r, t)$. Then

$$\frac{d}{dt} u^k(x_t, r, t) = \frac{\partial u^k}{\partial t}(x_t, r, t) + \frac{\partial u^k}{\partial x^i}(x_t, r, t)\, \dot{x}^i(t) + \frac{\partial u^k}{\partial r}(x_t, r, t)\, \dot{r}(t). \tag{20}$$

In RMF, $r$ is a fixed reference time independent of $t$, so $\dot{r}(t) = 0$ and the last term vanishes.

**Final coordinate expansion.** Combining (18) with (20) yields, for fixed $r$,

$$\left(\frac{D}{dt} u(x_t, r, t)\right)^k = \frac{\partial u^k}{\partial t} + \frac{\partial u^k}{\partial x^i}\, \dot{x}^i + \Gamma^k_{ij}(x_t)\, \dot{x}^i\, u^j. \tag{21}$$

If $r = r(t)$ is allowed to vary, the additional term remains:

$$\left(\frac{D}{dt} u(x_t, r, t)\right)^k = \frac{\partial u^k}{\partial t} + \frac{\partial u^k}{\partial x^i}\, \dot{x}^i + \frac{\partial u^k}{\partial r}\, \dot{r} + \Gamma^k_{ij}(x_t)\, \dot{x}^i\, u^j. \tag{22}$$

Equations (21)–(22) provide the standard "chain rule + Christoffel correction" expansion of $\nabla_{\dot{\gamma}(t)} u(x_t, r, t)$ in local coordinates.

## B. Local curvature-sensitive error bound for the practical surrogate

Let (M,g) be a smooth Riemannian manifold, and let $\gamma : [r, t] \to M$ be the interpolation path with $x_\tau = \gamma(\tau)$. Let

$$u(x_t, r, t) := v(x_t, t) - (t - r)\nabla_{\dot{\gamma}(t)} u(x_t, r, t) \tag{23}$$

be the exact RMF target in Eq. (6), and let

$$u_{\mathrm{gt}}(x_t, r, t) := v(x_t, t) - (t - r)\Big(\partial_t u + \partial_{x_t} u[\dot{x}_t]\Big) \tag{24}$$

be the practical target in Eq. (10) induced by the tangent-space JVP surrogate.

Assume that $\gamma([r,t])$ lies in a sufficiently small geodesically convex normal neighborhood $\Omega$ of $x_t$, and that on $\Omega$:

1. the norm of the Riemann curvature tensor satisfies $|R| \leq K$;

2. the path speed is bounded by $\|\dot{\gamma}(\tau)\|_g \leq V$;

3. the average-velocity field is bounded by $\|u(x_\tau, r, \tau)\|_g \leq U$.

**Lemma B.1.** *(Brewin, 2009). In normal coordinates centered at $x_t$, the Christoffel symbols satisfy the standard local estimate*

$$|\Gamma(x)| \leq C_0 K d(x, x_t), \qquad x \in \Omega. \tag{25}$$

**Proposition B.2.** *With $\Delta := t - r$,*

$$e_r := u(x_t, r, t) - u_{\mathrm{gt}}(x_t, r, t) \tag{26}$$

*satisfies*

$$|e_r|_g \leq C_0 U K V^2 \Delta^2. \tag{27}$$

For the proof, see Proof D.3.

**Error analysis of the practical surrogate.** The RMF identity is exact at the geometric level: average velocity is intrinsically defined by parallel transport, and curvature enters through the Levi–Civita connection and the covariant derivative along the interpolation path. The approximation arises only in the practical implementation, where the exact covariant derivative is replaced by a tangent-space JVP surrogate in a local log-map representation. In local coordinates, this discrepancy corresponds to omitting the Christoffel correction, and is therefore controlled by the local curvature, the path length, and the magnitude of the velocity field. Consequently, the surrogate is most reliable when the interpolation remains within a normal neighborhood where Exp/Log maps are smooth and the time interval is short. In highly curved regions or for long-range one-step transport, this approximation error may increase; in such cases, using shorter intervals or few-step generation can reduce the curvature-induced bias.

## C. Riemannian Manifolds with Closed-Form Geodesics

In this section, we describe the geodesics for several commonly used Riemannian manifolds, focusing on their closed-form expressions. We use the Geomstats Python package (Miolane et al., 2020a) to compute the geodesic paths, including the logarithmic (Log) and exponential (Exp) maps, which are essential for tasks such as interpolation and velocity field computation. Geomstats provides an efficient and user-friendly implementation for geometric operations on Riemannian manifolds, and we rely on its functionality to perform computations across various manifold domains such as Euclidean spaces, spheres, flat tori, and the special orthogonal group $\mathrm{SO}(3)$. The following subsections summarize the geodesic equations and the corresponding implementations for each manifold.

We first discuss the Euclidean space, followed by the sphere, flat torus, and finally $\mathrm{SO}(3)$. Each of these manifolds has specific properties that influence the geodesic behavior, and we present the relevant formulas and computational methods used in Geomstats to model these geodesics.

For a detailed introduction to geometric learning in Python with Geomstats, we refer to (Miolane et al., 2020b), which provides a comprehensive overview of the package's capabilities. For a more in-depth treatment of Riemannian geometry and its applications in machine learning, we recommend (Guigui et al., 2023), which includes both theoretical and implementation details.

### C.1. Euclidean Space

Euclidean Space is the simplest, flat example of a Riemannian manifold where the metric tensor is constant (the identity matrix in Cartesian coordinates). The exponential and logarithm maps in Euclidean space reduce to simple addition/subtraction as

$$\mathrm{Exp}_x(v) = x + v, \qquad \mathrm{Log}_x(y) = y - x. \tag{28}$$

These are exactly the "trivial" Exp/Log maps because the Euclidean Levi–Civita connection is flat, so parallel transport is identity and geodesics are straight lines.

### C.2. Sphere

We consider the unit sphere $\mathbb{S}^{d-1} = \{x \in \mathbb{R}^d : \|x\|_2 = 1\}$ equipped with the Riemannian metric induced by the ambient Euclidean space. Concretely, the tangent space at $x \in \mathbb{S}^{d-1}$ is

$$T_x\mathbb{S}^{d-1} = \{v \in \mathbb{R}^d : \langle x, v \rangle = 0\}, \tag{29}$$

and the metric inner product is the restriction of the Euclidean inner product,

$$\langle u, v \rangle_x := u^\top v, \qquad u, v \in T_x\mathbb{S}^{d-1}. \tag{30}$$

Let $\theta = \arccos(x^\top y) \in [0, \pi]$ denote the geodesic angle between $x, y \in \mathbb{S}^{d-1}$. The exponential map at $x$ and logarithm map at $x$ admit closed forms:

$$\mathrm{Exp}_x(v) = \cos(\|v\|)\, x + \sin(\|v\|)\, \frac{v}{\|v\|}, \qquad v \in T_x\mathbb{S}^{d-1}, \tag{31}$$

$$\mathrm{Log}_x(y) = \frac{\theta}{\sin\theta}\Big(y - (x^\top y)\, x\Big), \qquad y \in \mathbb{S}^{d-1},\ y \neq -x, \tag{32}$$

with $\mathrm{Log}_x(x) = 0$. Here $\|v\| = \sqrt{\langle v, v \rangle_x}$ is the norm induced by the metric. Under this geometry, geodesics are great-circle arcs and the geodesic distance is $d(x, y) = \theta$.

### C.3. Flat Torus

We consider the $N$-dimensional flat torus parameterized by angles

$$\mathbb{T}^N := [0, 2\pi)^N \quad \text{(with wrap-around)}, \tag{33}$$

equivalently $\mathbb{T}^N = \mathbb{R}^N/(2\pi\mathbb{Z})^N$. The tangent space is

$$T_x\mathbb{T}^N \cong \mathbb{R}^N, \tag{34}$$

and we use the constant (flat) metric inherited from $\mathbb{R}^N$, i.e.,

$$\langle u, v \rangle_x := u^\top v, \qquad u, v \in T_x\mathbb{T}^N. \tag{35}$$

The exponential map is addition followed by wrapping:

$$\mathrm{Exp}_x(u) = (x + u) \bmod 2\pi. \tag{36}$$

The logarithm map returns the shortest wrapped displacement (element-wise principal angle):

$$\mathrm{Log}_x(y) = \mathrm{atan2}\big(\sin(y - x),\, \cos(y - x)\big), \tag{37}$$

where $\sin(\cdot)$, $\cos(\cdot)$, and $\mathrm{atan2}(\cdot, \cdot)$ are applied element-wise.

**C.4.** $SO(3)$

We consider the special orthogonal group

$$SO(3) := \{R \in \mathbb{R}^{3\times3} : R^\top R = I, \ \det(R) = 1\}, \tag{38}$$

a 3D compact Lie group representing 3D rotations. The tangent space at $R$ is obtained by left translation of the Lie algebra

$$\mathfrak{so}(3) := \{\Omega \in \mathbb{R}^{3\times3} : \Omega^\top = -\Omega\}, \qquad T_R SO(3) = \{R\Omega : \Omega \in \mathfrak{so}(3)\}. \tag{39}$$

We endow $SO(3)$ with the standard bi-invariant Riemannian metric induced by the Frobenius inner product on the Lie algebra:

$$\langle R\Omega_1, R\Omega_2 \rangle_R := \frac{1}{2}\mathrm{tr}(\Omega_1^\top \Omega_2), \qquad \Omega_1, \Omega_2 \in \mathfrak{so}(3), \tag{40}$$

so that $\|R\Omega\|_R = \|\Omega\|_F / \sqrt{2}$ and the metric is invariant under left/right multiplication.

**Exponential and logarithm maps.** Under this metric, geodesics are given by one-parameter subgroups. The exponential map at $R$ is

$$\mathrm{Exp}_R(R\Omega) = R \exp(\Omega), \qquad \Omega \in \mathfrak{so}(3), \tag{41}$$

where $\exp(\cdot)$ is the matrix exponential. The logarithm map is defined on $R, S \in SO(3)$ (excluding the $\pi$-rotation ambiguity) by

$$\mathrm{Log}_R(S) = R \log(R^\top S), \tag{42}$$

where $\log(\cdot)$ is the matrix logarithm taking values in $\mathfrak{so}(3)$ (typically the principal logarithm).

**Axis–angle closed forms.** Let $Q := R^\top S \in SO(3)$ and define the relative rotation angle

$$\theta := \arccos\left(\frac{\mathrm{tr}(Q) - 1}{2}\right) \in [0, \pi]. \tag{43}$$

For $\theta \in (0, \pi)$, the principal logarithm admits the closed form

$$\log(Q) = \frac{\theta}{2\sin\theta}(Q - Q^\top) \in \mathfrak{so}(3), \tag{44}$$

hence

$$\mathrm{Log}_R(S) = R\frac{\theta}{2\sin\theta}(Q - Q^\top). \tag{45}$$

Conversely, if $\Omega \in \mathfrak{so}(3)$ and $\theta := \sqrt{\frac{1}{2}\mathrm{tr}(\Omega^\top\Omega)}$, then Rodrigues' formula gives

$$\exp(\Omega) = I + \frac{\sin\theta}{\theta}\Omega + \frac{1 - \cos\theta}{\theta^2}\Omega^2, \tag{46}$$

and therefore

$$\mathrm{Exp}_R(R\Omega) = R\left(I + \frac{\sin\theta}{\theta}\Omega + \frac{1 - \cos\theta}{\theta^2}\Omega^2\right). \tag{47}$$

At $\theta = 0$, use the limits $\frac{\sin\theta}{\theta} \to 1$ and $\frac{1-\cos\theta}{\theta^2} \to \frac{1}{2}$.

**Remark on ambiguity.** When $\theta = \pi$ (a 180° rotation), the logarithm is not unique; in practice one uses the principal branch when available or applies a consistent tie-breaking rule.

# D. Proof

### D.1. Proof of Proposition 3.2.

Recall that the RMF objective is

$$\mathcal{L}_{\text{RMF}} = \mathbb{E}_{x_0,x_1,r,t}\left[\left\|u_\theta(x_t,r,t) - u_{\text{gt}}(x_t,r,t)\right\|_g^2\right]. \tag{48}$$

Using the decomposition of the ground-truth target

$$u_{\text{gt}}(x_t,r,t) = v(x_t,t) - (t-r)\,\nabla_{\dot\gamma(t)}u(x_t,r,t), \tag{49}$$

we obtain

$$\mathcal{L}_{\text{RMF}} = \mathbb{E}_{x_0,x_1,r,t}\left[\left\|u_\theta(x_t,r,t) - v(x_t,t) + (t-r)\nabla_{\dot\gamma(t)}u(x_t,r,t)\right\|_g^2\right]. \tag{50}$$

Let $a := u_\theta(x_t,r,t) - v(x_t,t)$ and $b := (t-r)\nabla_{\dot\gamma(t)}u(x_t,r,t)$. Expanding the squared $g$-norm via $\|a+b\|_g^2 = \|a\|_g^2 + 2\langle a,b\rangle_g + \|b\|_g^2$, we have

$$\mathcal{L}_{\text{RMF}} = \mathbb{E}\left[\left\|u_\theta(x_t,r,t) - v(x_t,t)\right\|_g^2\right] + 2\,\mathbb{E}\left[\left\langle u_\theta(x_t,r,t) - v(x_t,t),\,(t-r)\nabla_{\dot\gamma(t)}u(x_t,r,t)\right\rangle_g\right]$$
$$+ \mathbb{E}\left[\left\|(t-r)\nabla_{\dot\gamma(t)}u(x_t,r,t)\right\|_g^2\right]. \tag{51}$$

Separating the cross term yields

$$\mathcal{L}_{\text{RMF}} = \mathbb{E}\left[\left\|u_\theta(x_t,r,t) - v(x_t,t)\right\|_g^2\right] + 2\,\mathbb{E}\left[\left\langle u_\theta(x_t,r,t),\,(t-r)\nabla_{\dot\gamma(t)}u(x_t,r,t)\right\rangle_g\right] + C, \tag{52}$$

where

$$C := \mathbb{E}\left[\left\|(t-r)\nabla_{\dot\gamma(t)}u(x_t,r,t)\right\|_g^2\right] - 2\,\mathbb{E}\left[\left\langle v(x_t,t),\,(t-r)\nabla_{\dot\gamma(t)}u(x_t,r,t)\right\rangle_g\right] \tag{53}$$

does not depend on the network parameters $\theta$. Therefore, when optimizing w.r.t. $\theta$, the gradient satisfies $\nabla_\theta\mathcal{L}_{\text{RMF}} = \nabla_\theta\big(\mathbb{E}[\|u_\theta - v\|_g^2] + 2\,\mathbb{E}[\langle u_\theta,(t-r)\nabla_{\dot\gamma}u\rangle_g]\big)$, and the constant term $C$ can be ignored.

### D.2. Derivation of Eq. (8).

Let $v_0 := \text{Log}_{x_1}(x_0) \in T_{x_1}\mathcal{M}$ and define the (constant-speed) geodesic

$$\gamma(s) := \text{Exp}_{x_1}(s\,v_0), \qquad s \in [0,1]. \tag{54}$$

Then Eq. (7) is simply a reparameterization of this geodesic:

$$x_t = \text{Exp}_{x_1}\big(\kappa(t)\,v_0\big) = \gamma(\kappa(t)). \tag{55}$$

By the chain rule,

$$\dot{x}_t = \frac{\text{d}}{\text{d}t}\gamma(\kappa(t)) = \kappa'(t)\,\gamma'(\kappa(t)). \tag{56}$$

It remains to express $\gamma'(s)$ in terms of $\text{Log}_{\gamma(s)}(x_1)$. Along the geodesic $\gamma$, the velocity satisfies $\gamma'(s) = \mathcal{P}_{0\to s}^\gamma(v_0)$, where $\mathcal{P}_{0\to s}^\gamma : T_{x_1}\mathcal{M} \to T_{\gamma(s)}\mathcal{M}$ denotes parallel transport. Moreover, within a normal neighborhood (so that the logarithmic map is single-valued), we have the standard identity

$$\text{Log}_{\gamma(s)}(x_1) = -s\,\mathcal{P}_{0\to s}^\gamma(v_0), \tag{57}$$

i.e., the displacement from $\gamma(s)$ back to $x_1$ is opposite to the forward direction of the geodesic and has magnitude equal to the geodesic distance. Combining $\gamma'(s) = \mathcal{P}_{0\to s}^\gamma(v_0)$ with Eq. (57) yields

$$\gamma'(s) = -\frac{1}{s}\,\text{Log}_{\gamma(s)}(x_1), \qquad s > 0. \tag{58}$$

Substituting $s = \kappa(t)$ into Eq. (56) gives

$$\dot{x}_t = \kappa'(t)\,\gamma'(\kappa(t)) = \kappa'(t)\left(-\frac{1}{\kappa(t)}\,\mathrm{Log}_{x_t}(x_1)\right) = -\frac{\kappa'(t)}{\kappa(t)}\,\mathrm{Log}_{x_t}(x_1), \tag{59}$$

which is exactly Eq. (8).

**Special case $\kappa(t) = 1 - t$.** When $\kappa(t) = 1 - t$, we have $\kappa'(t) = -1$, and thus

$$\dot{x}_t = -\frac{-1}{1-t}\,\mathrm{Log}_{x_t}(x_1) = \frac{1}{1-t}\,\mathrm{Log}_{x_t}(x_1), \tag{60}$$

which matches Eq. (9).

### D.3. Proof of Proposition B.2.

In the practical implementation, Eq. (10) replaces the exact covariant derivative by the tangent-space JVP surrogate

$$\partial_t u + \partial_{x_t} u[\dot{x}_t]. \tag{61}$$

Appendix A shows that the exact covariant derivative equals the Euclidean chain-rule term plus the Christoffel correction:

$$\left(\frac{D}{dt}u\right)^k = \frac{\partial u^k}{\partial t} + \frac{\partial u^k}{\partial x^i}\dot{x}^i + \Gamma^k_{ij}(x)\dot{x}^i u^j. \tag{62}$$

Hence the derivative-level discrepancy is controlled by the omitted connection term, so locally

$$\left|\nabla_{\dot{\gamma}(t)}u - (\partial_t u + \partial_{x_t} u[\dot{x}_t])\right|_g \leq |\Gamma(x)||\dot{\gamma}(t)|_g |u|_g. \tag{63}$$

Using the standard normal-coordinate estimate in Lemma B.1 and the path-length bound

$$d(x_r, x_t) \leq \int_r^t |\dot{\gamma}(s)|_g ds \leq V(t-r) = V\Delta, \tag{64}$$

we obtain

$$\left|\nabla_{\dot{\gamma}(t)}u - (\partial_t u + \partial_{x_t} u[\dot{x}_t])\right|_g \leq C_0 U K V^2 \Delta. \tag{65}$$

Multiplying by the prefactor $(t-r) = \Delta$ in the RMF identity yields

$$|e_r|_g \leq C_0 U K V^2 \Delta^2. \tag{66}$$

## E. Additional Experiments

We further evaluate RMF against recent MeanFlow variants, including $\alpha$-Flow (Zhang et al., 2026) and Improved Mean-Flow (Geng et al., 2026). We adapt these methods to Riemannian manifold generation as follows.

### E.1. $\alpha$-Flow

We adapt $\alpha$-Flow (Zhang et al., 2026), with the specific form as follows:

$$L_{\alpha RMF} = \mathop{\mathbb{E}}_{x_0, x_1, r, t}\left[\alpha^{-1} \cdot \|u_\theta(x_t, r, t) - [\alpha \cdot v(x_s, s) + (1-\alpha)\,\mathbf{sg}(u_\theta(x_s, r, s))]\|_g^2\right], \tag{67}$$

where $s = \alpha \cdot r + (1-\alpha) \cdot t$ and $x_s = \mathrm{Exp}_{x_1}(\kappa(s)\,\mathrm{Log}_{x_1} x_0)$.

### E.2. Improved MeanFlow (iMF)

We further adapt Improved MeanFlow (iMF) (Geng et al., 2026) to Riemannian manifolds, yielding *iRMF*. The resulting objective is

$$L_{iRMF} = \mathbb{E}_{x_0, x_1, r, t}\left[\|u_\theta(x_t, r, t) - v(x_t, t) + (t-r)\,\mathbf{sg}(\xi_t)\|_g^2\right], \tag{68}$$

$$\xi_t := u_\theta(x_t, t, t)\,\partial_{x_t} u_\theta + \partial_t u_\theta. \tag{69}$$

*Figure 5.* iRMF: Cosine Similarity about $\nabla\mathcal{L}_1(\theta)$ and $\nabla\mathcal{L}_2(\theta)$ with 0% r=t on Spherical Datasets.

Compared with RMF, we replace the trajectory velocity $\dot{x}_t$ (which is typically obtained from geodesic derivatives) by the network prediction $u_\theta(x_t, t, t)$. Empirically, this variant improves upon RMF.

Figure 5 reports the cosine similarity between the loss gradients of iRMF on the spherical dataset. We find that iRMF does not mitigate gradient conflicts; instead, it exhibits more frequent negative cosine similarities than RMF, indicating stronger gradient interference.

Table 8 further compares these methods under one-step sampling (1 NFE) using MMD. The gains of $\alpha$-Flow over RMF are marginal. iMF provides a modest improvement over RMF, but the overall improvement remains limited.

## F. Empirical Estimation of Training Runtime

Compared with RMF, RMF-MT introduces an additional cost during training due to the gradient-orthogonalization step used to mitigate conflicts between the decomposed objectives. Table 9 reports training throughput in iterations per second. RMF-MT remains efficient, incurring only a moderate slowdown relative to RMF (6.42 vs. 8.27 it/s), while still running substantially faster than the strong baseline G-LSD (Davis et al., 2026) (3.60 it/s). Overall, the added overhead of RMF-MT is modest and yields improved optimization stability and performance in exchange.

## G. Additional Results

In Figure 6, we present the cosine similarity between the gradients of the two losses across all datasets. The Spherical dataset has already been analyzed in Section 5.1, so we will not delve into further details here. On the torus datasets, we found that Glycine dataset has the most positive cosine similarity. Correspondingly, on glycine, RMF-MT shows the least improvement over RMF. This once again demonstrates that the benefit of RMF-MT is dataset-dependent.

In Figure 7, we conduct multiple sampling on Sphere, torus and SO(3) dataset and report MMD(↓). We can see that RMF achieves low MMD already at 1 step and remains relatively stable across different step counts. This indicates that RMF provides substantially better sample quality in the low-step regime, while RFM typically requires many iterative sampling steps to approach comparable performance.

In Table 10 and 11, we present the NLL scores of our method on the Spherical dataset and the Torus dataset. In Table 12, we present the KLD scores of our method on the Spherical dataset. We use the same testing protocol in (Chen & Lipman, 2024; Davis et al., 2026).

In Table 13 and 14, we clarify capacity vs. dataset scale below. For the hypersphere ablation studies, d=128 is already well beyond toy 2D/3D settings; the model size is only 732K to 860K parameters across d=3–128, with 50K samples in each case.

*Table 8.* MMD (↓) on the spherical dataset (mean over 5 runs with different random seeds). Best is **bold**; second best is underlined.

|  | Volcano | Earthquake | Flood | Fire |
|---|---|---|---|---|
| Dateset size | 827 | 6,120 | 4,875 | 12,809 |
| **RMF** | **0.092** | 0.042 | 0.068 | 0.042 |
| **RMF-MT** | 0.102 | **0.035** | **0.048** | **0.032** |
| $\alpha$RMF | 0.106 | 0.051 | 0.061 | 0.043 |
| iRMF | 0.108 | 0.039 | 0.060 | 0.037 |

*Table 9.* Iterations per second (↑) for different methods.

|  | RMF | RMF-MT | G-LSD (Davis et al., 2026) |
|---|---|---|---|
| iteration/second | **8.27** | 6.42 | 3.60 |

In Table 15, we can see that shrinking the model to 1/16 causes only mild degradation on several datasets, while more aggressive shrinking hurts harder datasets more. Thus capacity matters, but the pattern is not consistent with simple memorization.

In Table 16, for the MMD two-sample test, we compare 1000 generated and 1000 held-out test samples, and compute a permutation-test p-value with 100 permutations for the null hypothesis that both sets come from the same distribution. The p-values show a clear degradation trend as the training data decreases: most datasets are not statistically distinguishable at 100% data, several still pass at 10%, and all become distinguishable at 1%. Since we use 100 permutations, values of 0.01 indicate the minimum reported p-value resolution.

For memorization, we use a leave-one-out 1-Nearest Neighbor two-sample test (ideal Acc.=0.5); A/B denote train-generated/test-generated. We report 1-NN only on the sphere-type datasets, since reliable torus nearest-neighbor comparison would require a separate geodesic implementation. The 1-NN memorization diagnostic in Table 17 shows that the accuracy gap (A/B) between train-generated and test-generated is generally small and does not exhibit a consistent pattern of generated samples being substantially closer to the training set. Thus, the degradation in the low-data regime is not well explained by simple train-set memorization.

Overall, these results in Table 16 and Table 17 suggest that RMF retains substantial distribution-recovery ability in the 10% regime on several datasets, while showing clear degradation in the more extreme 1% regime, without a consistent signature of simple memorization.

*Table 10.* NLL(↓) on the Spherical Dataset. Standard deviation is estimated over 5 runs.

|  | Volcano | Earthquake | Flood | Fire |
|---|---|---|---|---|
| Dataset size | 827 | 6,120 | 4,875 | 12,809 |
| RDM | $-6.61 \pm 0.97$ | $-0.40 \pm 0.05$ | $0.43 \pm 0.07$ | $-1.38 \pm 0.05$ |
| RFM | $\mathbf{-7.93 \pm 1.67}$ | $-0.28 \pm 0.08$ | $0.42 \pm 0.05$ | $-1.86 \pm 0.11$ |
| G-LSD | $-4.96 \pm 0.68$ | $-0.93 \pm 0.01$ | $-0.38 \pm 0.33$ | $-2.14 \pm 0.42$ |
| G-PSD | $-3.50 \pm 0.22$ | $-0.63 \pm 0.13$ | $-0.76 \pm 0.13$ | $-2.48 \pm 0.71$ |
| G-ESD | $-4.49 \pm 0.20$ | $-0.67 \pm 0.08$ | $-0.88 \pm 0.38$ | $-2.29 \pm 0.08$ |
| RMF | $-3.73 \pm 0.41$ | $-1.08 \pm 0.09$ | $-0.72 \pm 0.11$ | $-2.24 \pm 0.30$ |
| RMF-MT | $-5.60 \pm 1.15$ | $\mathbf{-1.38 \pm 0.161}$ | $\mathbf{-0.907 \pm 0.11}$ | $\mathbf{-2.75 \pm 0.287}$ |

*Table 11.* NLL(↓) on the Torus Dataset. Standard deviation is estimated over 5 runs.

|  | General | Glycine | Proline | PrePro | RNA |
|---|---|---|---|---|---|
| Dataset size | 138,208 | 13,283 | 7,634 | 6,910 | 9,478 |
| RDM | $1.04 \pm 0.012$ | $1.97 \pm 0.012$ | $0.12 \pm 0.011$ | $1.24 \pm 0.004$ | $-3.70 \pm 0.592$ |
| RFM | $1.01 \pm 0.025$ | $\mathbf{1.90 \pm 0.055}$ | $0.15 \pm 0.027$ | $1.18 \pm 0.055$ | $\mathbf{-5.20 \pm 0.067}$ |
| G-LSD | $0.99 \pm 0.05$ | $1.99 \pm 0.02$ | $0.24 \pm 0.07$ | $1.11 \pm 0.02$ | $-4.15 \pm 0.09$ |
| G-PSD | $\mathbf{0.95 \pm 0.02}$ | $1.94 \pm 0.03$ | $\mathbf{0.08 \pm 0.04}$ | $1.10 \pm 0.04$ | $-4.40 \pm 0.13$ |
| G-ESD | $0.99 \pm 0.04$ | $1.95 \pm 0.01$ | $0.19 \pm 0.04$ | $1.10 \pm 0.02$ | $-4.61 \pm 0.07$ |
| RMF | $0.97 \pm 0.01$ | $1.97 \pm 0.01$ | $0.21 \pm 0.04$ | $\mathbf{1.02 \pm 0.04}$ | $-3.79 \pm 0.09$ |
| RMF-MT | $0.993 \pm 0.41$ | $2.04 \pm 0.11$ | $0.19 \pm 0.06$ | $1.084 \pm 0.022$ | $-4.68 \pm 0.21$ |

# H. Visualization

The visualization of generative proteins is presented in Figure 8. The visualization of the generation data for SO(3) is depicted in Figure 9.

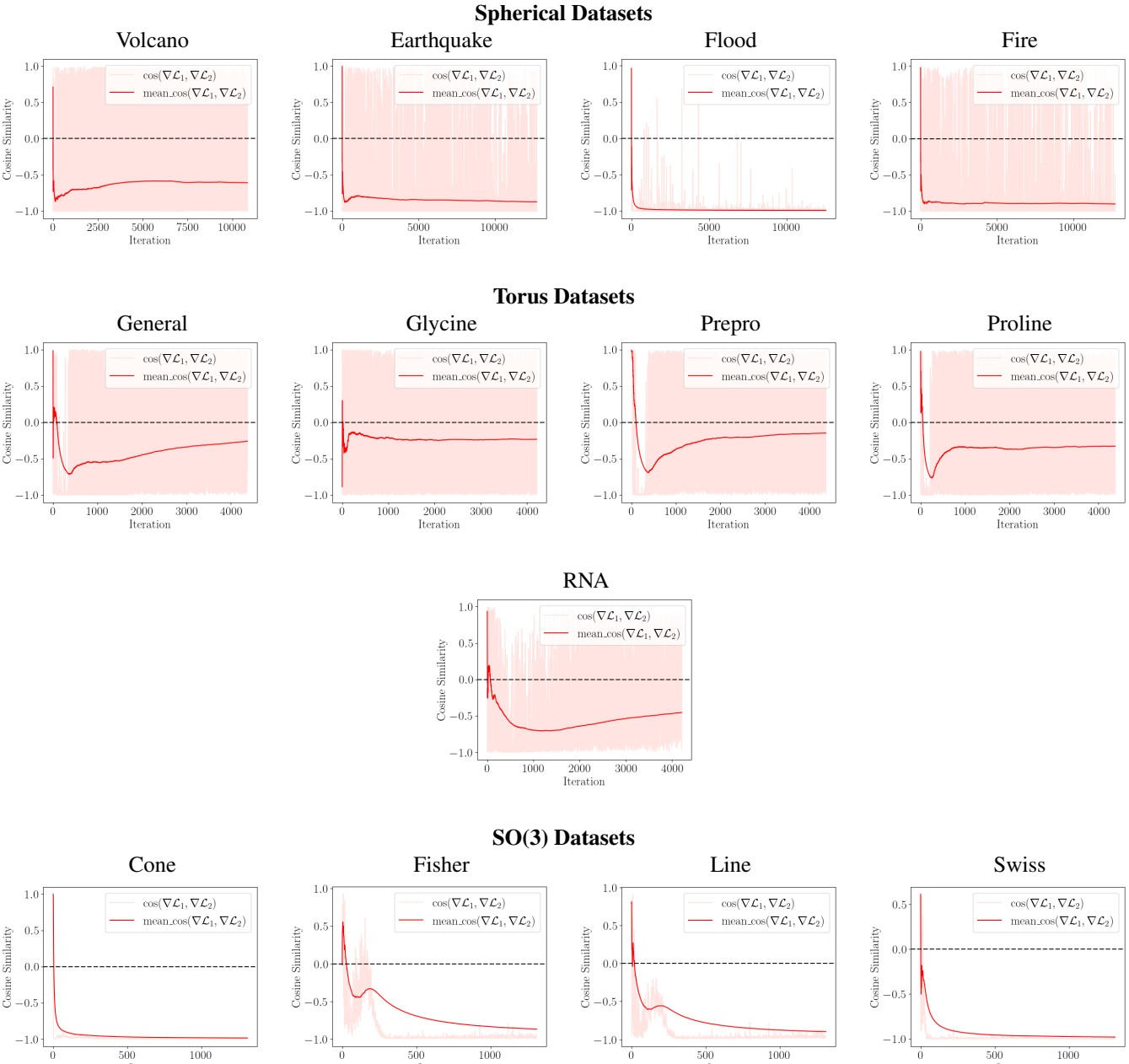

*Figure 6.* Cosine Similarity about $\nabla \mathcal{L}_1(\theta)$ and $\nabla \mathcal{L}_2(\theta)$ with 0% r=t

**All Datasets Multi-step Result**

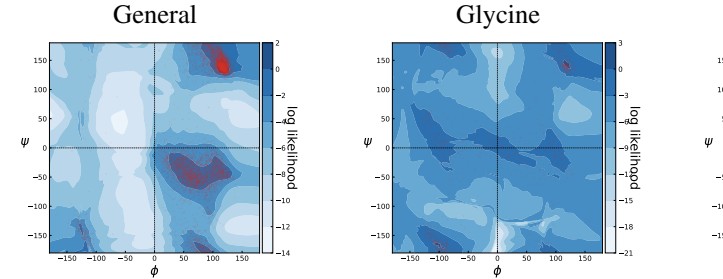

*Figure 7.* Multi-step MMD comparison between RFM and RMF across all datasets.

*Table 12.* KLD(↓) on the Spherical Dataset.

|              | Volcano | Earthquake | Flood | Fire   |
|--------------|---------|------------|-------|--------|
| Dataset size | 827     | 6,120      | 4,875 | 12,809 |
| RFM          | 472     | 43         | 49    | 39     |
| RCT          | 280     | 20         | 18    | **17** |
| RMF-MT       | **140** | **16**     | **17**| 19     |

General          Glycine          Proline          Prepro

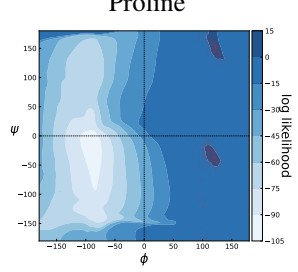
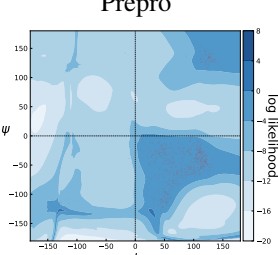

*Figure 8.* Ramachandran plots on the Torus Dataset.

# I. Additional Experimental Details

In Table 18 and Table 19, we present the hyperparameter settings of our model. The spherical, torus, SO(3), and CFG experiments are all summarized in these tables. The ablation study reuses the parameter configuration of the spherical model, with modifications limited solely to the input data.

*Table 13.* Dataset scale vs. network size in Ablation Studies. "data dimension $\times$ train samples" reports data dimension $\times$ number of training samples for each dataset; "network params" reports the number of RMF parameters used in that experiment.

| Data dim $S^N$ | 3 | 4 | 8 | 16 | 32 | 64 | 128 |
|---|---|---|---|---|---|---|---|
| data dim $\times$ train samples | 3$\times$50K | 4$\times$50K | 8$\times$50K | 16$\times$50K | 32$\times$50K | 64$\times$50K | 128$\times$50K |
| network params | 732K | 733K | 737K | 745K | 762K | 795K | 860K |

*Table 14.* Dataset scale vs. network size in other dataset. "D." is "data dimension". "T. S." is "train samples". "N. P." is "network params"

| Dataset | Fire | Flood | Earthquake | Volcano | General | PrePro | Glycine | Proline | RNA |
|---|---|---|---|---|---|---|---|---|---|
| D. $\times$ T. S. | 2$\times$12,809 | 2$\times$4,875 | 2$\times$6,120 | 2$\times$827 | 2$\times$138,208 | 2$\times$6,910 | 2$\times$13,283 | 2$\times$7,634 | 7$\times$9,478 |
| N. P. | 36M | 22.6M | 36M | 46M | 731K | 731K | 2.7M | 731K | 1.4M |

*Table 15.* RMF model downsizing ablation (MMD$\downarrow$). "base" denotes the original model; 1/16, 1/128, and 1/1024 denote reduced-size variants. "–" indicates experiments stopped once test performance deteriorated substantially.

| Dataset | Fire | Flood | Earthquake | Volcano | General | PrePro | Glycine | Proline | RNA |
|---|---|---|---|---|---|---|---|---|---|
| model size | 36M | 22.6M | 36M | 46M | 731K | 731K | 2.7M | 731K | 1.4M |
| base | 0.032 | 0.048 | 0.035 | 0.102 | 0.04 | 0.05 | 0.03 | 0.04 | 0.07 |
| 1/16 | 0.030 | 0.051 | 0.031 | 0.141 | 0.128 | 0.095 | 0.032 | 0.139 | 0.185 |
| 1/128 | 0.031 | 0.053 | 0.049 | 0.229 | - | 0.122 | 0.081 | - | - |
| 1/1024 | 0.068 | 0.097 | 0.072 | - | - | - | - | - | - |

*Table 16.* Low-data regime MMD two-sample evaluation. We report both MMD$\downarrow$ and permutation-test p-value$\uparrow$, computed by comparing 1000 generated samples with 1000 held-out test samples using 100 random permutations. Values of 0.01 indicate the minimum reported p-value resolution of the test.

| Dataset | Metric | Fire | Flood | Earthquake | Volcano | General | PrePro | Glycine | Proline | RNA |
|---|---|---|---|---|---|---|---|---|---|---|
| 100% | MMD | 0.032 | 0.048 | 0.035 | 0.102 | 0.04 | 0.05 | 0.03 | 0.04 | 0.07 |
| | p-value | 0.171 | 0.465 | 0.405 | 0.752 | 0.059 | 0.158 | 0.227 | 0.128 | 0.01 |
| 10% | MMD | 0.036 | 0.084 | 0.064 | 0.100 | 0.085 | 0.077 | 0.081 | 0.074 | 0.275 |
| | p-value | 0.107 | 0.049 | 0.089 | 0.108 | 0.01 | 0.01 | 0.326 | 0.01 | 0.01 |
| 1% | MMD | 0.066 | 0.085 | 0.133 | 0.225 | 0.110 | 0.125 | 0.127 | 0.167 | 0.281 |
| | p-value | 0.01 | 0.01 | 0.01 | 0.039 | 0.01 | 0.01 | 0.01 | 0.01 | 0.01 |

*Table 17.* Low-data regime leave-one-out 1-NN memorization diagnostic (accuracy $\rightarrow$ 0.5). A/B denote train-generated / test-generated comparisons, respectively. Each point is classified by the label of its nearest neighbor after excluding itself. Results are reported only on sphere-type datasets; torus datasets are omitted because reliable geodesic nearest-neighbor comparison would require a separate periodic-distance implementation.

| Training Data Fraction | Fire(A/B) | Flood(A/B) | Earthquake(A/B) | Volcano(A/B) |
|---|---|---|---|---|
| 100% | 0.7845/0.796 | 0.632/0.612 | 0.703/0.746 | 0.944/0.981 |
| 10% | 0.7935/0.793 | 0.673/0.657 | 0.735/0.735 | 0.948/0.982 |
| 1% | 0.8595/0.867 | 0.685/0.679 | 0.81/0.75 | 0.957/0.982 |

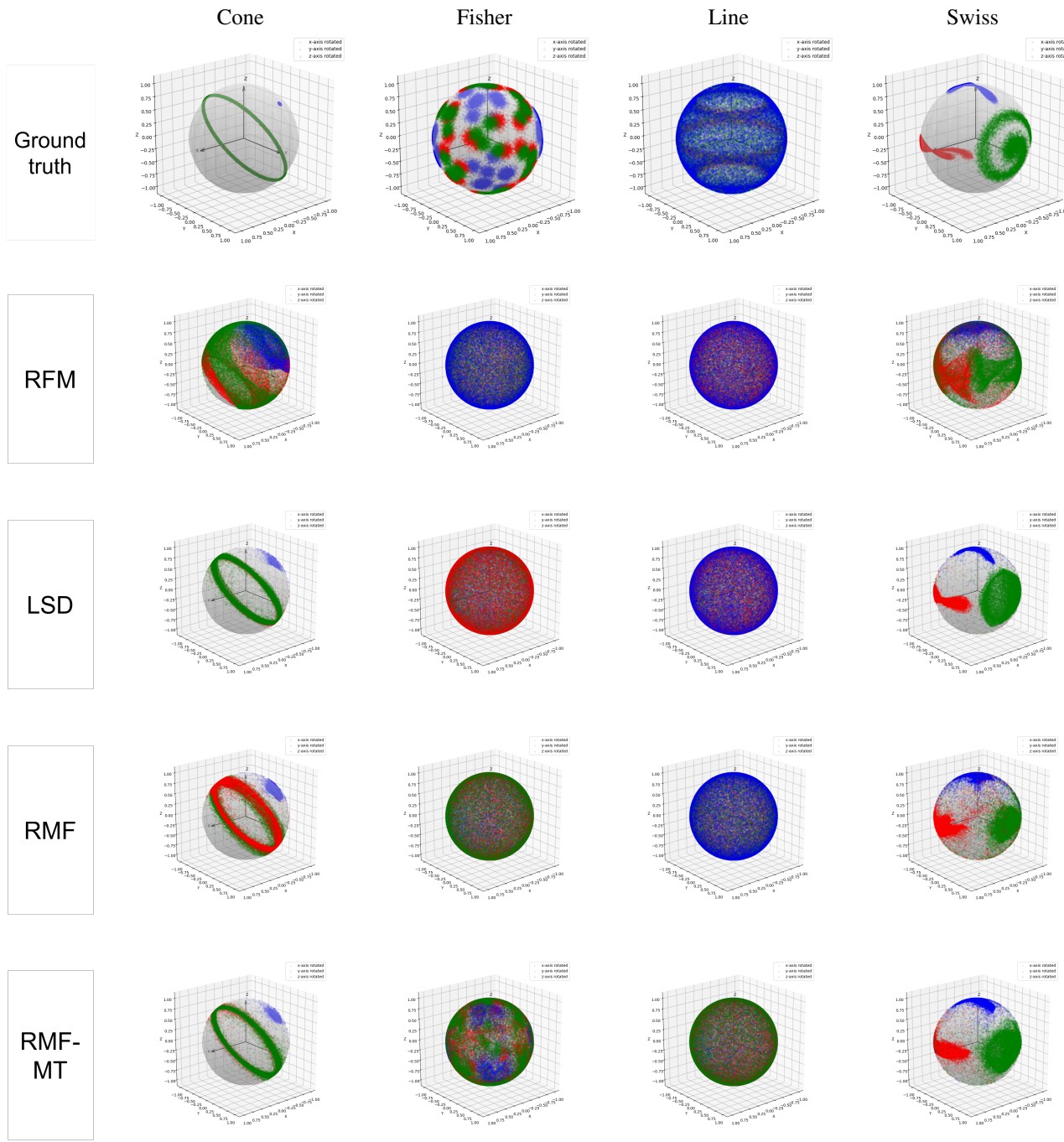

*Figure 9.* SO(3) one-step sample figure

*Table 18.* The Spherical and Torus experimental hyperparameters are as follows.

|  | Volcano | Earthquake | Flood | Fire | General | Glycine | Proline | PrePro | RNA |
|---|---|---|---|---|---|---|---|---|---|
| Dateset size | 827 | 6,120 | 4,875 | 12,809 | 138,208 | 13,283 | 7,634 | 6.910 | 9,478 |
| learning rate | 5e-4 | 5e-4 | 5e-4 | 5e-4 | 5e-4 | 5e-4 | 5e-4 | 5e-4 | 5e-4 |
| number of layers | 12 | 10 | 7 | 10 | 4 | 10 | 4 | 4 | 6 |
| hidden dimensions | 2048 | 2048 | 2048 | 2048 | 512 | 512 | 512 | 512 | 512 |
| r=t % | 75% | 75% | 75% | 75% | 75% | 75% | 75% | 75% | 75% |
| batch size | 8192 | 4096 | 8192 | 8192 | 4096 | 2048 | 2048 | 2048 | 2048 |
| input dim | 3 | 3 | 3 | 3 | 2 | 2 | 2 | 2 | 7 |
| epoch | 2000 | 700 | 700 | 600 | 5000 | 5000 | 5000 | 5000 | 2000 |
| optimizer | AdamW | | | | | | | | |
| lr schedule | CosineAnnealingLR | | | | | | | | |
| weight decay | 0.01 | | | | | | | | |
| (r,t) cond | (r, t) | | | | | | | | |

*Table 19.* The SO(3) experimental hyperparameters are as follows.

|  | Cone | Fisher | Line | Swiss Roll | CFG |
|---|---|---|---|---|---|
| Dateset size | 20K | 40K | 40K | 40K | 60k |
| learning rate | 5e-4 | 5e-4 | 5e-4 | 5e-4 | 5e-4 |
| number of layers | 4 | 4 | 4 | 8 | 4 |
| hidden dimensions | 512 | 512 | 512 | 1024 | 512 |
| r=t % | 10% | 10% | 10% | 10% | 10% |
| batch size | 1024 | 1024 | 1024 | 1024 | 1024 |
| input dim | 9 | 9 | 9 | 9 | 9 |
| epoch | 200 | 200 | 200 | 200 | 200 |
| optimizer | AdamW | | | | |
| lr schedule | CosineAnnealingLR | | | | |
| weight decay | 0.01 | | | | |
| (r,t) cond | (r, t) | | | | |

