# OpenReview forum: "Riemannian MeanFlow for One-Step Generation on Manifolds"
_ICML.cc/2026/Conference — ICML 2026 regular_

### Official Review · Reviewer_8fXA · 2026-03-02

**Soundness:** 3
**Presentation:** 3
**Significance:** 3
**Originality:** 2
**Overall Recommendation:** 3
**Confidence:** 4

**Summary:**

This work presents Riemannian MeanFlow to enable one-step generation on manifolds by extending the MeanFlow identity to location-dependent tangent spaces. By employing a log-map representation and parallel transport, the Riemannian MeanFlow avoids heavy geometric computations typically required in manifold flow matching. The framework treats its decomposed training objective as a multi-task optimization problem and applies PCGrad to resolve gradient conflicts. Experiments are conducted on spherical data, flat tori, and SO(3), demonstrating competitive one-step sampling performance with reduced sampling cost.

**Compliance With Llm Reviewing Policy:**

Affirmed.

**Final Justification:**

Overall, the rebuttal gave me a more positive impression of the work, but not enough for me to increase my recommendation at this time.

**Key Questions For Authors:**

1. How does RMF fundamentally differ from existing Riemannian Flow Matching or Riemannian diffusion approaches beyond replacing instantaneous velocity regression with average velocity regression?
2. Are additional assumptions required, such as geodesic completeness or bounded curvature? Are the guarantees local or global? Clarifying the exact geometric assumptions would improve rigor.

**Limitations:**

The core idea approximates trajectory-level information using an average-velocity identity. However, in highly curved manifolds, parallel transport effects may be non-negligible. Nonlinear geometry may amplify approximation errors. One-step generation may struggle when geodesic interpolation deviates strongly from true flow trajectories. The method does not explicitly analyze how curvature influences approximation fidelity.

**Strengths And Weaknesses:**

Strengths
The paper addresses a non-trivial issue in manifold generative modeling: defining and supervising average velocities when tangent spaces vary with location. The use of parallel transport to define an intrinsic average velocity is geometrically conceptually sound.

Weaknesses
While the paper provides an intrinsic reformulation of the MeanFlow identity on manifolds, the degree of conceptual novelty relative to existing Riemannian flow matching or diffusion-based frameworks remains somewhat unclear. The method appears to modify the supervision target, but it does not fully establish whether this leads to a fundamentally new dynamical formulation or is primarily a reparameterization of existing Riemannian flow training objectives. As a result, the theoretical distinction from prior manifold methods may be viewed as incremental rather than foundational.

---

> ### Author Rebuttal · Authors · 2026-03-31
>
> We appreciate that the reviewer recognizes the paper addresses a **non-trivial issue** in manifold generative modeling. We address the reviewer’s concerns as follows.
> ## 1. RMF learns finite-time transport rather than instantaneous dynamics.
>
> ---
> Regarding the conceptual distinction, we agree that RMF is **not** a fundamentally new probability-path family. Its main novelty is instead the **intrinsic object it learns**.
>
> Prior methods learn either **local instantaneous dynamics** or **shortcut/consistency-style flow maps**. For example, RFM and Riemannian diffusion methods model local vector fields, scores, or drifts, while RCM and GFM learn accelerated consistency or flow-map relations on manifolds. **RMF differs in that it learns a finite-interval average velocity field**.
>
> This is not a trivial change of supervision target. On a manifold, velocities along a trajectory lie in **different tangent spaces**, so a trajectory-level average is **not even well-defined** unless those vectors are first brought into a common space. Our key contribution is to resolve this geometric issue by defining average velocity **intrinsically via parallel transport**, and then deriving a **RMF identity** that connects this quantity to the instantaneous velocity through a covariant derivative.
>
> Therefore, RMF is best viewed as **a new intrinsic formulation of finite-interval transport learning on manifolds**. In this sense, it is more than a simple reparameterization of standard Riemannian flow training: the learned quantity itself is different, and making it well-defined on manifolds requires additional geometric structure.
>
> ## 2. Geometric assumptions and locality of the guarantees.
>
> ---
> Thank you for the important question. The manifolds in our experiments (e.g., spheres, tori, and $SO(3)$) are indeed **geodesically complete**, which makes them **tractable benchmark settings** with well-behaved geodesics and closed-form or manageable Exp/Log maps. However, the current derivation of RMF is primarily local: Prop. 3.1 relies on smoothness along the trajectory, and the practical implementation operates within a normal neighborhood where Exp/Log are well-defined and smooth. Hence, the present guarantees should be interpreted as **local rather than global**, and the derivation does not explicitly require **geodesic completeness** or a **global curvature bound**. That said, curvature can still influence the quality of the practical log-map-based approximation, so it affects approximation fidelity even if it is not an explicit theorem assumption. We will revise the paper to make this clearer.
> ## 3. Curvature-dependent limits of the one-step approximation.
>
> ---
> Thank you for this important comment. We agree that on **highly curved manifolds**, the gap between a **finite-interval average-velocity approximation** and the true trajectory dynamics can become larger; this is also **a broader challenge in Riemannian generative modeling**.
>
> Our method is nevertheless **intrinsic**: the average velocity is defined by **parallel transport** to a common tangent space, and the identity is written using the **covariant derivative** along the trajectory rather than a Euclidean derivative. Thus, curvature is not ignored; it enters through the connection, parallel transport, and the term $\nabla_{\dot{\gamma}(t)}u(x_t,r,t)$.
>
> That said, we agree that the current paper does not provide an explicit **curvature-dependent error bound**. The RMF identity is exact at the geometric level under sufficient smoothness; the approximation arises only in the practical implementation, where we use a **log-map tangent-space representation** and a neural approximation. This practical form is local and is most reliable when the trajectory stays in a region where Exp/Log are well behaved.
>
> We will clarify this point in the revision as follows:
> 1. We will explicitly state that RMF does **not** claim curvature-independent fidelity; its approximation is expected to be most accurate when the interpolation stays in regions where Exp/Log are well behaved and the local tangent-space approximation is reliable.
> 2. We will clarify that when curvature is large, or when geodesic interpolation departs substantially from the true probability-flow trajectory, the error of one-step generation can increase; this is a real limitation rather than something we intend to overlook.
> 3. Although we do not isolate curvature directly in the current paper, we will note that RMF still performs robustly across **spheres, tori, and $SO(3)$**, and scales favorably on higher-dimensional hyperspheres.
> 4. We will add this as an explicit limitation and discuss curvature-sensitive error analysis. We will also clarify that multi-step variants may be preferable in regimes where curvature-induced deviation is too strong for accurate one-step transport.

---

> > ### Author Rebuttal · Reviewer_8fXA · 2026-04-03
> >
> > Thank you for the response. While the rebuttal clarifies the distinctions from prior work as well as the geometric assumptions and locality of the guarantees, it does not fully resolve my concerns regarding the paper's core contributions and limitations.
> >
> > In particular, while the authors emphasize that the proposed method differs in learning a finite-interval average velocity field, this distinction does not yet convincingly establish a substantive conceptual advance over existing approaches such as RFM, RCM, or GFM, which already explore alternative supervision targets beyond instantaneous dynamics. Furthermore, it is unclear how sensitive the method is to geometric assumptions, and whether the additional structure required introduces limitations in more complex or less well-behaved manifold settings.
> > Therefore, while I view the paper more positively after the rebuttal, I do not believe the concerns are sufficiently resolved to justify raising my score to 4.

---

> > > ### Author Response · Authors · 2026-04-07
> > >
> > > ## 1. Main contribution
> > > We view RMF as a **formulation-level contribution rather than a fundamentally new** manifold probability-path family. Its core contribution is to make **finite-interval transport** the primary supervised object, and to show how this object can be made **intrinsic and well-posed on Riemannian manifolds** via parallel transport. In Euclidean space, such an interval-level quantity is natural, whereas on manifolds it is not directly available because velocities along a trajectory live in different tangent spaces. RMF addresses this geometric mismatch by transporting them to a common tangent space and defining an intrinsic average velocity there. Therefore, our claim is not that RMF is a foundational replacement for prior manifold generative frameworks, but that it provides **a previously unavailable finite-interval transport object made intrinsic and learnable on manifolds**.
> > > ## 2. Differences from existing methods
> > > | Method | Primary modeled object | Main emphasis |
> > > |---|---|---|
> > > | **RFM** | **Instantaneous vector field** | Learns local infinitesimal dynamics |
> > > | **RCM** | **Consistency / shortcut relation**  |  Emphasizes consistency across time for accelerated generation |
> > > | **GFM** | **Time-to-time transport / flow-map-style relation**| Emphasizes direct learning of transport relations across time |
> > > | **Ours (RMF)** | **Intrinsic finite-interval transport quantity**| Learns a finite-interval transport summary defined intrinsically on the manifold|
> > >
> > > RMF is centered on a **different primary object of learning**.
> > > ## 3. Parallel transport is not the approximation step
> > > Parallel transport under the Levi-Civita connection is a standard **exact intrinsic operation** for comparing tangent vectors across different tangent spaces, and in RMF it is used to define the average velocity itself. Therefore, it is **not** the approximation step at the level of the geometric definition. The actual approximation arises when the exact covariant derivative in the RMF identity is replaced by the practical surrogate **in a local-coordinate representation**. In local coordinates, the discrepancy can be interpreted as **neglecting the Christoffel correction** contained in the covariant derivative. Hence, the source of error in our framework is not parallel transport itself, but the **local surrogate used to approximate the covariant derivative**; the resulting discrepancy is controlled by **local curvature, time interval, and the magnitudes of the velocity fields**. In particular, the bias decreases with **the interval length**, so when curvature is high or long-range one-step transport becomes less reliable, it can be reduced in a principled way by **using shorter intervals and few-step generation**.
> > > ## 4. Proposition (Local curvature-sensitive error bound for the practical surrogate).
> > > Let $(M,g)$ be a smooth Riemannian manifold, and let $\gamma:[r,t]\to M$ be the interpolation path with $x_\tau=\gamma(\tau)$. Let
> > > $$
> > > u(x_t,r,t):=v(x_t,t)-(t-r)\nabla_{\dot\gamma(t)}u(x_t,r,t)
> > > $$
> > > be the exact RMF target in Eq. (6), and let
> > > $$u_{\mathrm{gt}}(x_t,r,t):=v(x_t,t)-(t-r)\Big(\partial_tu+\partial_{x_t}u[\dot x_t]\Big)
> > > $$
> > > be the practical target in Eq. (10) induced by the tangent-space JVP surrogate.
> > >
> > > Assume that $\gamma([r,t])$ lies in a sufficiently small geodesically convex normal neighborhood $\Omega$ of $x_t$, and that on $\Omega$:
> > >
> > > 1.	the norm of the Riemann curvature tensor satisfies $|R|\le K$;
> > > 2.	the path speed is bounded by $\|\dot\gamma(\tau)\|_g\le V$;
> > > 3.	the average-velocity field is bounded by $\|u(x_\tau,r,\tau)\|_g\le U$;
> > >
> > > ---
> > > **Lemma** [1]. In normal coordinates centered at $x_t$, the Christoffel symbols satisfy the standard local estimate
> > > $$|\Gamma(x)|\le C_0Kd(x,x_t),\qquad x\in\Omega.$$
> > >
> > > ---
> > > **Proposition**. With $\Delta:=t-r$,
> > > $$e_r:=u(x_t,r,t)-u_{\mathrm{gt}}(x_t,r,t)$$
> > > satisfies
> > > $$|e_r|_ g \le C_0UKV^2\Delta^2.$$
> > >
> > > ---
> > > ### Proof
> > > In the practical implementation, Eq. (10) replaces the exact covariant derivative by the tangent-space JVP surrogate
> > > $$\partial_tu+\partial_{x_t}u[\dot x_t].$$
> > > Appendix A shows that the exact covariant derivative equals the Euclidean chain-rule term plus the Christoffel correction:
> > > $$\left(\frac{D}{dt}u\right)^k=\frac{\partial u^k}{\partial t}+\frac{\partial u^k}{\partial x^i}\dot x^i+\Gamma^k_{ij}(x)\dot x^i u^j.$$
> > > Hence the derivative-level discrepancy is controlled by the omitted connection term, so locally
> > > $$\bigl|\nabla_{\dot\gamma(t)}u-(\partial_tu+\partial_{x_t}u[\dot x_t])\bigr|_ g\le|\Gamma(x)||\dot\gamma(t)|_ g|u|_ g.$$
> > > Using the standard normal-coordinate estimate in Lemma and the path-length bound
> > > $$d(x_r,x_t)\le \int_r^t |\dot\gamma(s)|_ gds \le V(t-r)= V\Delta,$$
> > > we obtain
> > > $$\bigl|\nabla_{\dot\gamma(t)}u-(\partial_tu+\partial_{x_t}u[\dot x_t])\bigr|_g\le C_0UKV^2 \Delta.$$
> > > Multiplying by the prefactor $(t-r)=\Delta$ yields the conclusion.
> > > $$|e_r|_g \le C_0UKV^2 \Delta^2.$$
> > >
> > > ---
> > > [1] Riemann Normal Coordinate expansions using Cadabra, 2009.

---

### Official Review · Reviewer_DKRA · 2026-03-13

**Soundness:** 3
**Presentation:** 4
**Significance:** 3
**Originality:** 3
**Overall Recommendation:** 4
**Confidence:** 3

**Summary:**

The paper extends MeanFlow for one step generation to the Riemannian setting where the manifold is known. After extending the meanflow identity, the paper derives a principled algorithm to produce a few step generator in the Riemannian setting. The method additionally improves on a naïve extension of MeanFlow by decomposing the MeanFlow loss into two conflicting objective, and improves upon naïve gradient descent by applying PCGrad, extending the preciously observed training instabilities for Euclidean MeanFlow models. Additionally, the method extends to conditional generation by adapting ideas from classifier-free guidance. The unconditional model is tested on standard Riemannian benchmarks, and the conditional model is tested on a synthetic benchmark.

**Compliance With Llm Reviewing Policy:**

Affirmed.

**Key Questions For Authors:**

- (Q1) Is this the first work exploring conditional generation on manifolds?

**Limitations:**

yes

**Strengths And Weaknesses:**

**Strengths:**

-	(S1) The paper is well written and easy to follow despite the mathematical content.
-	(S2) The problem of one-step generation is of high relevance, and MeanFlow is a particularly strong candidate, so extending it to the Riemannian setting is interesting.
-	(S3) The empirical result show strong performance against recently introduced competitive methods.

**Weaknesses:**

-	(W1) The novelty of the method is limited, as it can be seen as a straightforward extensions of MeanFlow to the non-Euclidean settings.
-	(W2) The applicability of the method is limited since it requires access to exponential and logarithm maps, which are only accessible in very simple cases.
-	(W3) The only performance metric is MMD. Previous work and baselines use KLD on the earth datasets [1] and test NLL on RNA torsion angles [2] to evaluate one step models on manifolds. It would improve the evaluation to add these metrics given that the baselines already use them.


[1] Davis, Oscar, Michael S. Albergo, Nicholas M. Boffi, Michael M. Bronstein, and Avishek Joey Bose. "Generalised Flow Maps for Few-Step Generative Modelling on Riemannian Manifolds." arXiv e-prints (2025): arXiv-2510.
[2] Cheng, Chaoran, Wang, Yusong, Chen, Yuxin, et al. Riemannian Consistency Model. arXiv preprint arXiv:2510.00983, 2025.

---

> ### Author Rebuttal · Authors · 2026-03-31
>
> We appreciate that the reviewer finds the paper **clear despite its mathematical content**, considers one-step generation in the Riemannian setting an **interesting and relevant problem**, and recognizes the paper’s **strong empirical performance against recent competitive methods**. We address the reviewer’s concerns as follows.
> ## W1. A nontrivial realization beyond direct extension.
>
> ---
> We acknowledge that our method is more appropriately viewed as an incremental advance rather than a fundamentally new paradigm. However, there is an important distinction between a **direct extension in principle** and a **practical realization on manifolds**. While RMF is indeed inspired by MeanFlow, its extension is **not straightforward**, because the core Euclidean quantity—**average velocity**—is no longer directly well-defined when velocities lie in **different tangent spaces**. Our main contribution is to make this quantity **well-defined, tractable, and optimizable** in the manifold setting. Moreover, we address gradient conflicts in the RMF objective, which improves the **stability of optimization** in practice. Finally, our framework naturally supports **conditional generation on manifolds**.
> ## W2. Applicability beyond only simple closed-form manifolds.
>
> ---
> We agree that RMF is most directly applicable on manifolds where exponential and logarithm maps are available in closed or efficient form. However, this does **not** limit the method to only **very simple cases**. Many **practically important domains** in geometric generation are defined on **spheres, tori**, and rotation/pose manifolds such as $SO(3)$ and $SE(3)$, or their product spaces. In these settings, Exp/Log maps are **standard computational primitives** rather than **restrictive assumptions**. Our formulation uses these maps precisely to avoid **coordinate-dependent Christoffel-symbol computations** and **explicit parallel transport**, yielding a **simpler and more practical** implementation in a common tangent space.
>
> That said, we agree that RMF does not yet cover arbitrary manifolds with no efficient geometric primitives. We view extension beyond manifolds with **tractable geodesic operations** as an important direction for future work, and we now make this scope more explicit.
> ## W3. NLL already included; KLD added for completeness.
>
> ---
> We thank the reviewer for this suggestion. We apologize for not including the KLD$\downarrow$ results for the Earth dataset in the main submission; we report them below for one-step generation.
> | method/dataset | Fire | Flood | Earthquake | volcano |
> | :----: | :----: | :----: | :----: |:----: |
> | RFM | 39 | 49 | 43 | 472 |
> | RCT | **17** | 18 | 20 | 280 |
> | RMF | 19 | **17** | **16** | **140** |
>
> These results are consistent with our main conclusions and show that **RMF remains competitive** against prior one-step baselines.
>
> In addition, NLL results were **already included** in the appendix at the time of submission: **Tables 8** reports Earth NLL, and **Table 9** reports Torus/RNA NLL.
>
> We highlighted MMD in the main paper because it directly measures the discrepancy between one-step generated samples and the test distribution, which matches our primary evaluation goal. We agree, however, that KLD and NLL are important complementary metrics and will present them more clearly in the revision.
> ## Question 1
>
> ---
> To the best of our knowledge at the time of submission (Jan. 29, 2026), this is the first work to incorporate classifier-free guidance into a Flow/Diffusion-style generative model on Riemannian manifolds.

---

> > ### Author Rebuttal · Reviewer_DKRA · 2026-04-01
> >
> > I thank the authors for the rebuttal. The addition of KLD makes the evaluation more complete. My concerns have been addressed and I maintain my positive assessment of the work.

---

> > > ### Author Response · Authors · 2026-04-02
> > >
> > > Thank you for your time and constructive feedback. We are glad the response addressed your concerns and appreciate your continued positive assessment.

---

### Official Review · Reviewer_vagm · 2026-03-24

**Soundness:** 3
**Presentation:** 1
**Significance:** 2
**Originality:** 3
**Overall Recommendation:** 3
**Confidence:** 4

**Summary:**

This work extends the one-step generation method, Mean Flow, to a manifold setting. The central contribution is the Riemannian Mean Flow Identity and the induced decomposed learning objective. The proposed approach is validated over three datasets, a spherical dataset, a torus dataset, and an SO(3) dataset, and is compared against several generative methods on smooth manifolds.

**Compliance With Llm Reviewing Policy:**

Affirmed.

**Final Justification:**

I appreciate the authors' responses and most of my concerns are resolved. I increased my score.

**Key Questions For Authors:**

see weaknesses.

**Limitations:**

Yes. The authors discussed the potential misuse of synthetic examples and over-reliance on generated outputs in high-stakes domains.

**Strengths And Weaknesses:**

Strength:

1. This work uplifts Mean Flow, originally developed in Euclidean space, to smooth manifolds, which is a natural piece to complete the Mean Flow theory.

2. The proposed method outperforms several baselines at NFE = 1 across multiple datasets.

Weaknesses:

1. This work is not appropriately motivated. Mathematically, it's a natural extension from Euclidean space to manifolds, but it discusses no practical needs of a single-step generative model on a manifold.

2. Many mathematical objects are not clearly defined. This makes it hard for the audience to process the technical part of the paper, especially for those less familiar with the paradigm of generative models on manifold.
Some examples are:

 2.1. Around the right column of line 79, what is the definition of a probability path on a manifold?

 2.2. How is the probability measure defined on a manifold? What are the assumptions on p0 and p1? What are the common design choices of p0 and the probability path?

 2.3 What is the definition of $\psi_t(x_0| x_1)$ in (3)?

3. The experiments are mostly on lower-dimension datasets, and some have very small sample sizes. The significance of the proposed methods is not fully conveyed through experiments on these datasets.
In particular, I think two critical pieces of evidence to support the significance of the proposed method are missiong from the paper: 1) examples where efficient Euclidean flow/diffusion sampling methods fail while the proposed method works. 2) examples where the high-quality samples are expensive to obtain via iterative sampling with models like RMF, but can be sampled with one or few steps with Riemannian Mean Flow. (I'd be happy if the authors can prove me wrong, but I feel that for datasets like 2D sphere, a standard flow/diffusion model could work well, although the samples might be off the sphere by a small $\epsilon$.)

4. Additionally, this work only compares the proposed method with the baseline when NFE = 1. It would be important to verify how much sample quality it sacrifices for faster computation. The simplest way is to evaluate the quality of samples generated with more steps.

---

> ### Author Rebuttal · Authors · 2026-03-31
>
> We appreciate that the reviewer recognizes RMF as **a natural completion** of the MeanFlow framework and the method’s **strong 1-NFE performance** across multiple datasets. We address the reviewer’s concerns as follows.
> ## 1. Practical need for one-step generation on manifolds.
>
> ---
> Thanks for the helpful comment. We agree that the motivation should be stated more explicitly. Our general goal is to address the **sampling speed bottleneck** in generation on manifolds. A representative application is **robotic grasping**, where 6DoF grasping poses lie on $SE(3)$ and fast candidate generation is important for the following real-time trajectory planning. To make this practical motivation explicit, we conducted a large-scale $SE(3)$ grasp-pose experiment over 7800 objects (Urain, J. et al., ICRA23). Evalution results below (**success rate%**) show that RMF is much stronger in the low-step regime. This directly supports the practical benefit of RMF in fast generation.
>
> | steps | 1 | 2 | 3 | 4 | 5 | 6 | 7 |
> | :----: | :----: | :----: | :----: |:----: |:----: |:----: |:----: |
> | RFM | 3.2 | 23 | 38 | 59 | 80 | 82 | 88 |
> | GFM | 70 | 85 | 86 | **93** | **93** | 92 | **95** |
> | RMF | **80** | **88** | **90** | **93** | 91 | **94** | **95** |
>
> ## 2. Definitions of the mathematical objects.
>
> ---
> Thanks for indicating this. We clarify those definitions in the following and will revise the main paper.
> ### 2.1 Probability paths on a manifold
> Following RFM, probability paths are defined through conditional paths:
> $p_t(x)=\int p_t(x|x_1)q(x_1)d\mathrm{vol}{x_1},$
> where $q(x_1)=p_1$, $p_t(x|x_1)$ are conditional paths, and $d\mathrm{vol}$ is the Riemannian volume measure. The path satisfies
> $p_0(\cdot\mid x_1)=p_0,\quad p_1(\cdot\mid x_1)= p_1.$
>
> ### 2.2 Measure, assumptions on $p_0,p_1$, and design choices
> All densities are defined with respect to the **Riemannian volume measure** (equivalently, the Borel probability measure induced by the manifold structure). Here, $p_1$ is the data distribution on $M$, and $p_0$ is a uniform distribution on $M$. The **main assumption** is that both are **valid probability distributions on the manifold**, and that the interpolation path is well defined on the manifolds.
>
> A common **design choice** for $p_0$ is uniform distribution. For example, on the sphere, one can sample a Gaussian vector in the ambient space and normalize it, rather than sampling angular coordinates independently. The probability path is chosen by conditional geodesic interpolation, as in RFM.
>
> ### 2.3 $\psi_t(x_0|x_1)$ in Eq. (3)
> We clarify that $\psi_t(x_0|x_1)$ denotes the geodesic interpolation flow between $x_0$ and $x_1$. $\psi_t(x_0|x_1)$ is the point obtained by moving from $x_1$ toward $x_0$ along the geodesic according to $\kappa(t)$.
>
> ---
> ## 3. Experimental significance and comparison beyond 1 NFE.
> Thanks for the insightful comment. First, regarding **the use of lower-dimensional and small datasets**: the datasets in our paper are standard benchmark datasets for manifold generative modeling, and were chosen to enable direct comparison with SOTA methods. The above additional robotic grasping dataset is large and contains 2000 parallel-jaw grasps for 7800 objects from 262 categories, totalling 15.6M grasps.
>
> (1) Euclidean flow
> **Our paper already includes a high-dimensional hypersphere ablation with Euclidean Meanflow.** In **Tab. 5**, Euclidean MeanFlow (EMF) degrades substantially as dimension increases, with MMD rising from 0.193 at $d=3$ to 0.357 at $d=128$, while RMF remains stable and achieves 0.026 at $d=128$. Besides, EMF need over 400 epoches for convergence at $d=128$, while RMF merely needs 6 epoches. This shows that the issue is not merely a small off-manifold deviation $\epsilon$, but a more fundamental limitation of Euclidean parameterizations for manifold-supported data.
>
> (2) obtain high-quality samples
> Thanks for the insightful comment. We conduct multiple sampling on the Earth dataset and report MMD ($\downarrow$) as following:
>
> | steps | 1 | 2 | 4 | 8 | 16 | 32 | 64 | 128 |
> | :----: | :----: | :----: | :----: |:----: |:----: |:----: |:----: |:----: |
> | RFM | 0.377 | 0.240 | 0.098 | 0.028 | 0.026 | 0.016 | 0.024 | 0.027 |
> | RMF | 0.032 | 0.023 | 0.032 | 0.026 | 0.018 | 0.024 | 0.025 | 0.026 |
>
> Results on other datasets are in https://anonymous.4open.science/r/figures-D552/multi-step.png
>
> These results directly support the key point: RFM needs multiple solver steps to reach high quality, while RMF already achieves comparable quality in one or more steps.
>
> ---
> ## 4. Sample quality sacrifice.
> Thanks for the valuable comment. The result above show that the quality gap is very small: RMF already achieves strong performance at 1–2 steps, and additional steps bring only limited improvement. In contrast, RFM requires substantially more NFEs before reaching comparable quality. We will incorporate these multi-step comparisons into the revised paper to make this trade-off explicit.

---

> > ### Author Rebuttal · Reviewer_vagm · 2026-04-02
> >
> > Thanks for the rebuttal.
> >
> > 1. The author lists a potential application of one-step generation on manifolds. This is definitely an improvement but I don't think this makes the paper strongly motivated. Could the author provide 2 or 3 more works to strengthen the argument that this is a critical problem to solve?
> >
> > 2. Thanks for educating me on the mathematical set up of generative models on manifolds. I'm curious why $p_0$ is fixed to be a uniform distribution. I'm wondering if there is a theoretical constraint and merely a common practical choice. I think to match literatures in flow and diffusion models, $p_0$ should be allowed to be a broader set of measures unless there is some intrinsic theoretical constraints. A more general $p_0$ would open up more applications like data-to-data generation, like flows in Euclidean space.
> >
> > Could the author be more precise on `the main assumption is that both are valid probability distributions on the manifold`? Also, does this contradict the earlier statement that `$p_0$ is a uniform distribution on $M$`?  If I pick finite number of points on a sphere and assign uniform mass among them, I think it will still be a well-defined measure under the topology induced by the sphere. However, I doubt that it will work with any data distributions.
> >
> > I would guess the needed assumptions should be on the existence of density and its support, like the assumptions in standard rectified flows.
> >
> > Also, could the author clarify that if  `$p_0$ is a uniform distribution on $M$` is a design choice or a theoretical assumption? The author said `A common design choice for $p_0$ is uniform distribution. ` Meanwhile, `all densities are defined with respect to the Riemannian volume measure (equivalently, the Borel probability measure induced by the manifold structure). Here, $p_1$ is the data distribution on $M$, and  $p_0$ is a uniform distribution on $M$. `
> >
> >
> > 3. and 4. Thanks for updating the experiments. It partially resolves for concern. I appreciate the experiments showing how Euclidean mean flow converges slowly.
> >
> > Can the authors also compare the number of parameters in the network against the size of data set in terms of dimension x number of samples? I'm not sure if $d=128$ should be considered as high-dimension but I now understood the lower-dimension datasets are actually standard benchmark.  However, my concern is that if the capacity of neural networks is significantly beyond than the size of data, there is a risk of memorization.
> >
> > Is it possible to downscale the neural nets and see how it performs?
> >
> > Is it possible to set up experiments with synthetic data under low-data regime and run some statistical test to on generated samples to assess if they recover the original distributions?

---

> > > ### Author Response · Authors · 2026-04-06
> > >
> > > ## 1. Practical motivation
> > > Thanks for the follow-up comment. We agree that **one application is not enough**. More broadly, **sampling efficiency is a recurring bottleneck in manifold-valued generation**. Besides robotic grasping, this also appears in **3D molecule/protein generation**, where SE(3)-aware/equivariant models are followed by **large-scale screening, validation, or downstream optimization [1–3], so iterative sampling cost directly affects practical usability**; and in **6DoF grasp generation**, where SE(3)-equivariant diffusion/flow models and explicit **grasp-sampling acceleration have already** been studied [4–7]. Therefore, **one-step/low-step manifold generation addresses a broader practical need**.
> > >
> > > Refs: [1] protein backbone FM (ICLR’24); [2] 3D molecule diffusion (ICML’22); [3] InvDesFlow-AL (2025); [4] SE(3)-DiffusionFields (ICRA’23); [5] EquiGraspFlow (CoRL’24); [6] faster 6-DoF grasp sampling (IROS’24); [7] EvolvingGrasp (ICCV’25).
> > >
> > > ## 2. Clarification on $p_0$
> > > Thanks for this important comment. $p_0$ **is not theoretically required to be uniform**. In our paper, uniform $p_0$ is only a **design choice** for simplicity. The intended assumption is that $p_0$ and $p_1$ **admit densities w.r.t. the Riemannian volume measure**, and that the interpolation path is well defined on their supports. A finitely supported uniform measure on the sphere is indeed a **valid Borel probability measure**, but it is generally **not absolutely continuous** w.r.t. the Riemannian volume measure, so it lies **outside** the present formulation. Hence, **uniform $p_0$ is a design choice, while the theoretical assumption is that $p_0$ and $p_1$ admit densities w.r.t. the Riemannian volume measure.**
> > >
> > > ## 3. Capacity, downsizing, and low-data regime
> > > Thanks for these suggestions. We clarify **capacity vs. dataset scale** below. Full tables are in https://anonymous.4open.science/r/figures-D552/Tables.png
> > >
> > > For the hypersphere experiments, d=128 is already **well beyond toy 2D/3D settings**; the model size is only **732K to 860K** parameters across d=3–128, with **50K samples** in each case. We view this only as a **rough sanity check**; the more direct evidence comes from the ablations below.
> > > ### 3.1 Downscaling
> > > Shrinking the model to **1/16**  causes only **mild degradation** on several datasets, while more aggressive shrinking hurts harder datasets more. Thus **capacity matters**, but the pattern is **not consistent with simple memorization**.
> > >
> > > |dataset|Fire|Earthquake|Glycine|
> > > |:----:|:----:|:----:|:----:|
> > > |base size|0.032|0.035|0.03|
> > > |1/16|0.030|0.031|0.032|
> > > |1/128|0.031|0.049|0.081
> > > ### 3.2 Low-data regime with statistical tests
> > > Following the reviewer’s suggestion, we added two diagnostics.
> > >
> > > **Setup**. For the **MMD two-sample test**, we compare **1000 generated** and **1000 held-out test** samples, and compute a **permutation-test p-value with 100 permutations** for the **null hypothesis** that both sets come from the same distribution. For memorization, we use a **leave-one-out 1-Nearest Neighbor two-sample test** (ideal Acc.=0.5); **A/B** denote **train-generated/test-generated**. We report 1-NN only on the **sphere-type datasets**, since reliable torus nearest-neighbor comparison would require a separate geodesic implementation.
> > >
> > > **MMD permutation test**. The **p-values** show a clear degradation trend as the training data decreases: most datasets are **not statistically distinguishable at 100%** data, several still pass at **10**%, and **all** become distinguishable at **1**%. Since we use **100** permutations, values of 0.01 indicate the minimum reported p-value resolution. The corresponding MMD values are provided in the anonymous link.
> > >
> > > |Data fraction|Fire|Flood|Earthquake|Volcano|General|PrePro|Glycine|Proline|RNA|
> > > |:----:|:----:|:----:|:----:|:----:|:----:|:----:|:----:|:----:|:----:|
> > > |100%|0.171|0.465|0.405|0.752|0.059|0.158|0.227|0.128|0.01|
> > > |10%|0.107|0.049|0.089|0.108|0.01|0.01|0.326|0.01|0.01|
> > > |1%|0.01|0.01|0.01|0.039|0.01|0.01|0.01|0.01|0.01|
> > >
> > > **The 1-NN memorization diagnostic** below shows that the **accuracy gap (A/B)** between **train-generated and test-generated is generally small** and does **not** exhibit a consistent pattern of generated samples being substantially closer to the training set. Thus, the degradation in the low-data regime is **not well explained by simple train-set memorization**.
> > >
> > > |Data fraction|Fire|Flood|Earthquake|Volcano|
> > > |:----:|:----:|:----:|:----:|:----:|
> > > |100%|0.785/0.796|0.632/0.612|0.703/0.746|0.944/0.981|
> > > |10%|0.794/0.793|0.673/0.657|0.735/0.735|0.948/0.982|
> > > |1%|0.86/0.867|0.685/0.679|0.81/0.75|0.957/0.982|
> > >
> > > Overall, these results suggest that RMF retains **substantial distribution-recovery ability** in the **10**% regime on several datasets, while showing **clear degradation** in the more extreme **1**% regime, **without a consistent signature of simple memorization**.

---

### Decision · Program_Chairs · 2026-04-30

**Decision:**

Accept (regular)

**Comment:**

The paper makes interesting contributions to generative models on (known) manifolds. The reviewers did not reach a consensus: some considered the paper interesting, while others found it too difficult to read. I strongly recommend that the authors take this feedback to heart and revise the paper to be more easily accessible to a broader audience. However, in my own reading of the paper, I found the work to be sufficiently valuable to the community that I recommend acceptance.